# Neural Experts: Mixture of Experts for Implicit Neural Representations

**Yizhak Ben-Shabat**[*]
Roblox, The Australian National University
`sitzikbs@gmail.com`

**Chamin Hewa Koneputugodage**[*]
The Australian National University
`chamin.hewa@anu.edu.au`

**Sameera Ramasinghe**
Amazon, Australia
`sameera.ramasinghe@adelaide.edu.au`

**Stephen Gould**
The Australian National University
`stephen.gould@anu.edu.au`

## Abstract

Implicit neural representations (INRs) have proven effective in various tasks including image, shape, audio, and video reconstruction. These INRs typically learn the implicit field from sampled input points. This is often done using a single network for the entire domain, imposing many global constraints on a single function. In this paper, we propose a mixture of experts (MoE) implicit neural representation approach that enables learning local piece-wise continuous functions that simultaneously learns to subdivide the domain and fit it locally. We show that incorporating a mixture of experts architecture into existing INR formulations provides a boost in speed, accuracy, and memory requirements. Additionally, we introduce novel conditioning and pretraining methods for the gating network that improves convergence to the desired solution. We evaluate the effectiveness of our approach on multiple reconstruction tasks, including surface reconstruction, image reconstruction, and audio signal reconstruction and show improved performance compared to non-MoE methods. Code is available at our project page <https://sitzikbs.github.io/neural-experts-projectpage/>.

## 1 Introduction

Implicit neural representations (INRs) are a type of neural network that can encode a wide range of signal types, including images, videos, 3D models, and audio [31, 27, 5, 39, 22]. Instead of storing discretely sampled signals (e.g., on a uniform grid), INRs approximate signals using a continuous function defined by a neural network. Given an input coordinate, the network is optimized to estimate the signal value at that coordinate. These neural representation networks have gained popularity because they can effectively fit even complex or high-dimensional signals for various tasks, including 3D reconstruction [31, 27, 2, 3, 14, 5], novel-view synthesis [28, 4], neural rendering [43], and pixel alignment [34].

Current neural representation models, while powerful, are often limited by their architecture, which typically involves multi-layer perceptrons (MLPs). This design has two inherent drawbacks. The first is a parallelization and scale limitation: since INRs represented by a MLP process each coordinate by the whole network, all parameters have to contribute to the output of every point in the domain, making parameter optimization difficult. The second is a locality limitation: a desirable property of INRs is for features to change rapidly (to allow modelling of sharp boundaries) which does not arise naturally in optimized MLPs due to spectral bias [41]. Consequently, pure-MLP networks inhibit the ability to scale, fit the signal, and perform local operations on signals represented by the network.

---

[*]Equal contribution.

38th Conference on Neural Information Processing Systems (NeurIPS 2024).

Recent advances in large language models (LLMs) have shown that facilitating scaling is crucial for performance [17, 45, 42]. In LLMs, this is often done using a mixture of experts (MoE) architecture [11, 16, 30, 38, 21]. MoEs provide inherent parallelizability, enabling efficient computation across different computation hardware. Additionally, MoEs inherently subdivide the input data, allowing the network to focus on distinct regions or features for more effective and specialized learning. While MoEs have proven their merit in various tasks (e.g., normal estimation [6], multiclass classification [8], and LLMs), they have so far been overlooked for INRs.

In this paper we introduce Neural Experts, a novel mixture of experts architecture for implicit neural representations (MoE INRs). This architecture can be applied to any existing MLP-based INR with enhanced levels of control over the network. Specifically, MoE INRs offer the flexibility to either explicitly control locality or allow it to be learned implicitly while also enabling new levels of parallelization. One intuitive explanation for the effectiveness of MoEs for INRs is that while traditional INRs approximate using continuous functions, MoEs can naturally represent using piecewise continuous functions, facilitated by the gating function of the manager network which has a global perspective of the signal throughout training. This effectively partitions the input space into regions of expertise for individual experts.

We demonstrate the effectiveness of our proposed Neural Experts approach on a variety of applications including the representation of images, audio, and 3D shapes. Our work takes an important step towards making neural representations more flexible and effective for a wide range of tasks.

Specifically, the key contributions of this work are:

- Introducing Mixture of Experts for Implicit Neural Representations, which simultaneously subdivides the domain, reconstructs signals, and enables parallelization.

- Deriving a novel manager architecture and initialization that enable domain subdivision without ground truth.

- Demonstrating the effectiveness of our approach on a diverse set of applications in image, audio, and 3D surface reconstruction, showing improved reconstruction performance with fewer parameters compared to non-MoE methods.

## 2   Related Work

**Implicit Neural Representations.** Implicit Neural Representations (INRs) are continuous function approximators often based on multilayer perceptrons (MLPs). Their continuous nature benefits many reconstruction tasks but is particularly advantageous when dealing with irregularly sampled signals, such as point clouds [31, 2, 14]. However, standard MLPs have been shown to perform poorly when fitting signals with high frequencies [41, 39]. A common method to address this is to apply frequency encoding in the MLP, where frequencies are explicitly provided to the network. This can either be done by Fourier feature encoding [28, 41] or by using activation functions such as sinusoidal [39], Gaussian [32], wavelet [35] and sinc [37] functions. However these approaches are sensitive to the frequency bandwidth at initialization, and lack control of the bandwidth [5, 22]. Thus improvements include supplying a large bandwidth and reducing it over optimization using regularization [5] or explicit frequency decomposition [48, 22].

Another approach is to specialise subsets of the parameters for certain regions of the domain. This can be done by specialising parameters for each cell in a grid of the domain [7], which can be improved by making it hierarchical [40], adaptive [26] or operate on Laplacian pyramids [36]. Another method is to have a set of weights that get chosen based on the input's spatial position with a predetermined pattern [15, 23], or by hashing [29]. However the former is inefficient parameter-wise for general use, where the complexity of the signal could be localized to specific regions, and the latter does not specialise the parameters to important regions naturally.

Our method, on the other hand, allows the locally operating parameters (expert subnetworks), to change their local region during training. This is done by their joint optimization with the manager network that subdivides the domain.

**Mixture of Experts** The mixture of experts framework [16, 18, 30] has shown to be very effective for different tasks. Its strengths stems from its architectural design that subdivides a network into multiple experts and a manager network (also referred to as gating or routing network), and its loss

that encourages the manager to give higher weighting to better performing experts. This allows the network to subdivide the input space based on the task's loss and encourages different experts to specialize in reconstructing specific signals over that space.

Many works have focused on the MoE framework including formulating the experts as Gaussian processes [44], adding experts sequentially [1], introducing hierarchy [47], and ensembles-like formulation [13]. Recent advancements have shown that it is crucial for scaling up large language models [17, 45, 42]. In particular, since the capacity of a network is limited by the number of parameters it has, Shazeer et al. [38] propose MoE layers as a general purpose neural component to drastically scale parameters without a proportional increase in computation. This allows for improved parallelization and scaling [21], but cannot use the original MoE loss and requires a balancing loss to ensure load balancing and sparsity. Various works propose to use this layer within INRs [49, 46], however their use of the layers do not allow for sharp boundaries like the original MoE formulation.

Given the advantages of MoEs, we propose to apply similar principals to INRs. Unlike a straightforward extension, we will show that training an INR-based manager requires careful initialization and conditioning for improved performance.

It is important to note that the MoE framework can only work for INRs within a task that gives a loss per instance (which the MoE framework then changes to a weighted sum of losses over experts). Thus while the MoE framework can be used for many INR tasks (such as signal representation and reconstruction), it cannot be used for NeRF. Rebain et al. [33] show a NeRF-specific solution that significantly departs from the MoE-framework.

# 3 Mixture of Experts for Implicit Neural Representations

We present a novel approach for INRs that overcomes limitations by naively applying the mixture of experts method of Jacobs et al. [16, 30] in the reconstruction setting. Specifically, direct application fails to produce satisfactory results because the manager network is unable to fully take advantage of the supervisory signal provided to the experts. Our approach adapts the MoE architecture to share contextual information between manager and experts, and introduces a mechanism for pre-training the manager that avoids bad local minima and balances expert assignment.

**Preliminaries and Vanilla MoE INRs.** The original formulation of Jacobs et al. [16, 30] encourages different sub-networks, called experts, to specialize on a subset of the data. This is done using a manager network that outputs a vector of probabilities for the selection of each expert $q_i$. Let the parameters be $\theta_m$ for the manager, and $\theta_e^{(i)}$ for the parameters for the $i$-th expert.

As a straightforward extension and baseline, we implemented a naive extension (vanilla) of the Mixture of Experts to INRs as depicted in Figure 1b. In this vanilla formulation the manager's expert selection probability $q = \Phi_m(x; \theta_m)$ and the final reconstructed signal output $\Phi(x) = \Phi^{(j)}(x; \theta^{(j)})$, where $j = \text{argmax}\{q_1, \ldots, q_N\}$, are modeled as MLPs. However, for implicit neural representations, this formulation does not capitalize on the clear benefits of subdividing the input space since it is susceptible to converging to local minima. Therefore we propose the neural experts formulation.

**Neural Experts.** Our proposed architecture, illustrated in Figure 1c, is composed of two modules: the Manager and Experts modules. The experts act as the signal reconstructors while the manager acts as a routing function, indicating which expert should be used for each input sample $x$. For brevity, in the sequel, we will omit parameters from the MLP function $\Phi(x; \theta)$ and denote it simply as $\Phi(x)$.

1. Experts: The experts module is subdivided into two components, both modeled using multi-layered perceptrons (MLP). The encoder, parameterized by $\theta_e^E$, processes input samples $x$ to produce an intermediate representation $\Phi_e^E(x)$, here the subscript refers to the module (experts), while the superscript denotes the encoder component within it (Encoder). This representation is then fed into the next component, which includes $N_{\text{experts}}$ different experts. Each expert has its own architecture and is parametrized by its own set of parameters $\theta_e^{(i)}$, to output a reconstructed signal per expert $\Phi_e^{(i)}(\Phi_e^E(x))$.

2. Manager: The manager module is also composed of two components, both also modeled using MLPs. The manager encoder, parametrized by $\theta_m^E$, receives the input samples $x$ and outputs an intermediate representation $\Phi_m^E(x)$. A key component of our approach is to condition the manager using the signal. To do that, we concatenate the expert encoder

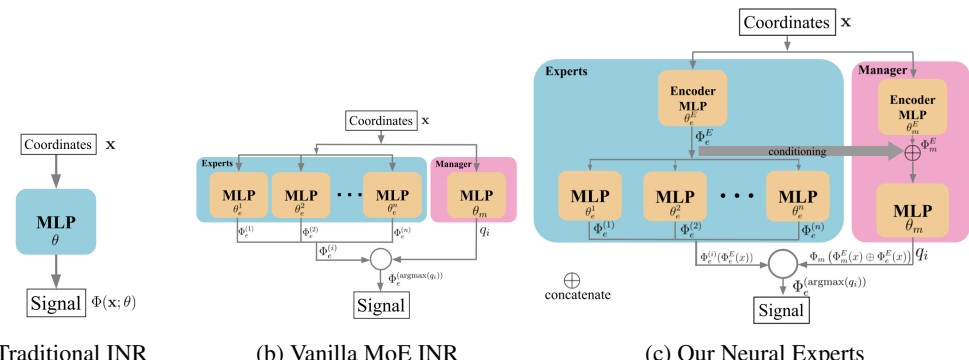



(a) Traditional INR      (b) Vanilla MoE INR      (c) Our Neural Experts



Figure 1: Illustration comparing between INR architectures for (a) traditional INR, (b) Vanilla MoE INR and (c) the proposed Neural Experts. Two key elements of our approach include the conditioning and pretraining of the manager that improve signal reconstruction with fewer parameters.

output with the manager encoder output and feed that into the next MLP. This results in the manager's final output $q = \{q_i \mid i = 1, \ldots N_{\text{experts}}\}$, which provides the selection criteria of each expert. More formally, let $\Phi_{\text{m}}^E(x) \oplus \Phi_{\text{e}}^E(x)$ be the concatenation of outputs from the manager encoder and expert encoder, respectively. Then

$$q(x) = \Phi_{\text{m}} \left( \Phi_{\text{m}}^E(x) \oplus \Phi_{\text{e}}^E(x) \right) \tag{1}$$

is the selection probability vector for all experts.

Finally, let $j = \text{argmax}\{q_1, \ldots, q_N\}$ denote the index of the expert with largest selection probability. Then the reconstructed signal is given by

$$\Phi(x) = \Phi_{\text{e}}^{(j)}(\Phi_{\text{e}}^E(x)) \tag{2}$$

It is important to note that the expert outputs are chosen based on the manager's prediction before expert computation is executed. This approach facilitates parallelization of expert computations (at inference) by routing samples exclusively to the most suitable expert.

**Manager pretraining.** For INRs, the initialization is crucial for performance [39, 2, 5]. Since the manager network is essentially another INR we propose to pretrain it to output a random uniform expert assignment. For that, we generate the random uniform assignment $y_{\text{seg}}$ as ground truth and train the manager using a cross entropy loss for each input coordinate,

$$L_{\text{seg}}(x) = CE(q(x; \theta_{\text{m}}^E, \theta_{\text{m}}, \theta_{\text{e}}^E), y_{\text{seg}}(x)) \tag{3}$$

The segmentation loss is then averaged over all input samples.

This loss provides two main benefits. First, it shapes the distribution of $q$ at initialization and second, it balances the assignment for the different experts. This balance is crucial for the MoE formulation as it helps prevent starving some experts while over-populating others.

This loss can also be used during the training process when a ground truth segmentation is available and a segment-per-expert is desired. Note that, as can be expected, in some of our experiments this yielded lower reconstruction performance (additional constraint to satisfy) and therefore is not used in the reconstruction pipeline, however, having each expert correspond to a semantic entity may have benefits for downstream tasks.

Note that manager pretraining is independent of the signal and therefore acts as an initialization.

**MoE loss.** After pretraining the manager, we reconstruct the signals by training both manager and experts (including encoders) for a set amount $t_{\text{all}}$ of the training iterations using the MoE reconstruction loss:

$$L_{\text{Recon-MoE}}(x) = \frac{1}{N_{\text{experts}}} \sum_{i=1}^{N_{\text{experts}}} q_i \cdot \left( \Phi_{\text{e}}^{(i)} \left( \Phi_{\text{e}}^E(x) \right) - y_{\text{gt}}(x) \right)^2 \tag{4}$$

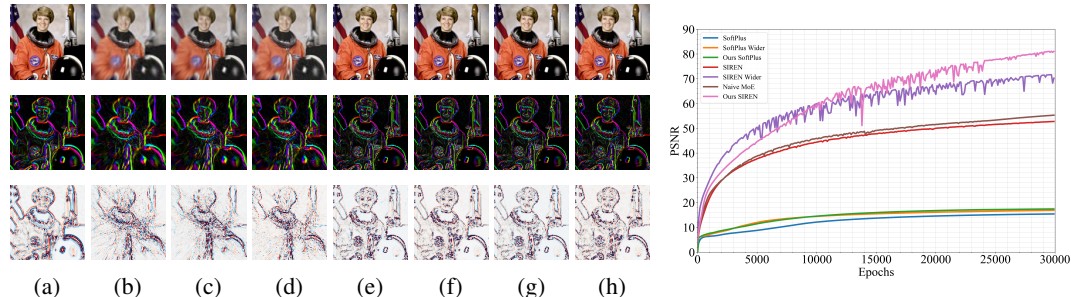

| (a) | (b) | (c) | (d) | (e) | (f) | (g) | (h) |

Figure 2: **Image reconstruction.** Qualitative (left) and quantitative (right) results. Showing image reconstruction (top), gradients (middle), and laplacian (bottom) for (a) GT, (b) SoftPlus, (c) SoftPlus Wider, (d) Our SoftPlus MoE, (e) SIREN, (f) SIREN Wider, (g) Naive MoE, and (h) Our SIREN MoE. The quantitative results (right) report PSNR as training progresses and show that our Neural Experts architecture with Sine activations outperforms all baselines.

The reconstruction loss is then averaged over all input samples.

During the final $t_e$ number of training iterations, we train only the experts in order to relax some of the instability caused by concurrently changing the manager assignments. In this stage, the parameters $\theta_m, \theta_m^E$ and $\theta_e^E$ are frozen and only the $\theta_e^{(i)}$ get updated.

## 4 Neural Experts Experimental Results

### 4.1 Image reconstruction

We conduct a comprehensive evaluation of image reconstruction on the full Kodak dataset [12] (24 images) in Table 1. The reconstruction task aims to encode the input signal (image in this case) into the neural network. At test time, the coordinates are fed into the network and the reconstruced image is compared to the ground truth.

The results show that our approach provides a significant boost in performance compared to other baselines. In this experiment we compare the proposed Neural Experts approach to prominent MLP architectures with $SoftPlus$ [2, 3], $SoftPlus + FF$ [41], $Sine$ [39] and $FINER$ [24] activations. For a fair comparison, we also implement a 'Wider' variant of these architectures that matches the INR's width with the combined number of elements over all experts. This results in an increased number of parameters for the 'Wider' variations. The results, presented in Figure 2 show that the proposed Neural Experts approach provides a significant boost in performance despite having fewer parameters than the 'Wider' baselines.

In Table 2 we present additional experiments focused on the Sine activation function. We report results across several architecture variants, including a smaller version of our Neural Experts model, a baseline Vanilla MoE INR, and a version of the baseline model enhanced with random pretraining. The results reveal several key trends: first, our method shows substantial improvement over the naive Vanilla MoE model; second, random pretraining significantly benefits the Vanilla MoE model's performance; and finally, our smaller version of the Neural Experts model remains competitive. Additional qualitative and quantitative results (including comparisons to hybrid methods like InstantNGP [29]), as well as specific architecture and training details, are available in the supplemental material.

### 4.2 Audio signal reconstruction

**Comparison to baselines.** To demonstrate the versatility of our Neural Experts, we follow SIREN [39] and show its application to audio signal reconstruction. We evaluate the performance on raw audio waveforms of varying length clips of music (*bach*), single person speech (*counting*), and two person speech (*two speakers*). Table 3 shows the mean-squared error of the converged models. The results show that the proposed MoE architecture is able to outperform the MLP-based representations with over 40% fewer parameters. Our Neural Experts approach was able to converge quicker to a representation which can be played with barely noticeable distortion.

| Activation | Method | Params. | PSNR | | SSIM | | LPIPS | |
|---|---|---|---|---|---|---|---|---|
| | | | Mean | Std | Mean | Std | Mean | Std |
| Softplus | Base | 99.8k | 19.51 | 2.95 | 0.7158 | 0.1106 | 4.08e-1 | 8.33e-2 |
| Softplus | Wider | 642.8k | 20.91 | 3.12 | 0.7798 | 0.0899 | 3.38e-1 | 8.44e-2 |
| SoftPlus | Ours | 366.0k | 20.62 | 3.12 | 0.7628 | 0.0946 | 3.53e-1 | 8.47e-2 |
| Softplus + FF | Base | 99.8k | 28.97 | 3.30 | 0.9433 | 0.0193 | 7.59e-2 | 1.70e-2 |
| Softplus + FF | Wider | 642.8k | 29.48 | 3.91 | 0.9436 | 0.0245 | 7.74e-2 | 2.30e-2 |
| SoftPlus + FF | Ours | 366.0k | 31.66 | 3.16 | 0.9652 | 0.0180 | 4.13e-2 | 1.57e-2 |
| Sine | Base | 99.8k | 57.23 | 2.46 | 0.9991 | 0.0005 | 5.78e-4 | 5.09e-4 |
| Sine | Wider | 642.8k | 77.50 | 5.32 | 0.9996 | 0.0005 | 3.08e-4 | 3.04e-4 |
| Sine | Ours | 366.0k | 89.35 | 7.10 | **0.9997** | 0.0004 | 2.49e-4 | 2.93e-4 |
| FINER | Base | 99.8k | 58.08 | 3.04 | 0.9991 | 0.0005 | 7.47e-4 | 1.31e-3 |
| FINER | Wider | 642.8k | 80.32 | 5.40 | 0.9996 | 0.0004 | 2.49e-4 | 2.46e-4 |
| FINER | Ours | 366.0k | **90.84** | 8.14 | **0.9997** | 0.0004 | **2.46e-4** | 2.48e-4 |

Table 1: **Image reconstruction with different activations.** Reporting performance of various activations on the Kodak dataset [12]. Our approach outperforms the baselines (Base, Wider).

| Method | Params. | PSNR | | SSIM | | LPIPS | |
|---|---|---|---|---|---|---|---|
| | | Mean | Std | Mean | Std | Mean | Std |
| Base | 99.8k | 57.23 | 2.46 | 0.9991 | 0.0005 | 5.78e-4 | 5.09e-4 |
| Ours small | 98.7k | 63.42 | 7.09 | 0.9992 | 0.0007 | 9.96e-4 | 2.40e-3 |
| Wider | 642.8k | 77.50 | 5.32 | 0.9996 | 0.0005 | 3.08e-4 | 3.04e-4 |
| Vanilla MoE | 349.6k | 62.98 | 4.16 | 0.9993 | 0.0005 | 4.53e-4 | 3.88e-4 |
| Vanilla MoE + Random Pretraining | 349.6k | 74.28 | 7.36 | 0.9996 | 0.0004 | 3.05e-4 | 2.88e-4 |
| Ours | 366.0k | **89.35** | 7.10 | **0.9997** | 0.0004 | **2.49e-4** | 2.93e-4 |

Table 2: **Image reconstruction architecture ablations with Sine activation.** Comparing our method to baselines on the Kodak dataset [12] with Sine activations. Our method shows substantial improvement over the naive Vanilla MoE model, random pretraining significantly enhances the Vanilla MoE model's performance, and our smaller Neural Experts model remains competitive.

All architectures were trained for 30k iterations, however noticeable performance gaps are already evident at 10k iterations (including manager pretraining).

In Figure 3 we show qualitative results of the reconstructed audio signal of the *two speakers* signal compared to the ground truth. The results show that while reconstruct the signal quite well, the proposed approach yields lower errors. For more audio reconstruction visualizations please see the supplemental material.

**Using speaker identity as auxilary supervision.** An interesting observation can be made from Figure 4 where we compare the reconstruction of the *two speakers* example with and without speaker identity segmentation information. Here, segmentation refers to labeling the time span for each one of the different speakers. The results show that the manager is able to learn this segmentation well, essentially allocating an expert per speaker (and an expert to background noise and gaps). Surprisingly, despite the additional segmentation constraint, this yields slightly better MSE score of 0.15 (compared to 0.16 without segmentation), however we witnessed that the segmentation supervision introduces slower convergence time (achieving comparable results only after $\sim 15k$ iterations. This result highlights possible future applications involving editing capabilities of INRs.

| Architecture | Bach | Counting | Two Speakers | # parameters |
|---|---|---|---|---|
| SIREN | 0.71 | 36.6 | 2.06 | $\sim 100K$ |
| SIREN Wider | 0.49 | 15.9 | 0.51 | $\sim 642K$ |
| Our SIREN MoE | **0.12** | **1.72** | **0.16** | $\sim 365K$ |

Table 3: **Audio reconstruction**. Reporting mean squared error (MSE),divided by $10^{-4}$ for brevity. The results show that the proposed Neural Experts method is able to significantly outperform non-MoE architectures while utilizing fewer parameters.

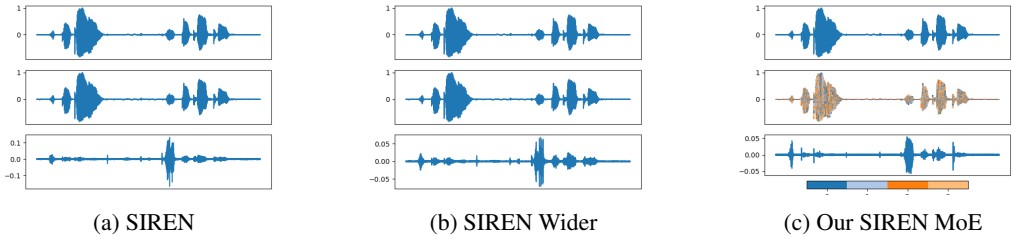

| (a) SIREN | (b) SIREN Wider | (c) Our SIREN MoE |

Figure 3: **Audio reconstruction visualization.** *Two speakers* audio reconstruction is presented. Within each waveform block, the rows represent the ground truth, reconstruction, and error visualization from top to bottom. For our Neural Experts we color code the different experts on the reconstructed waveform.

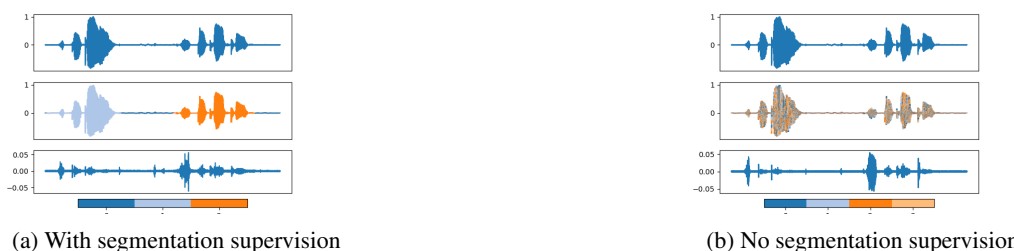

| (a) With segmentation supervision | (b) No segmentation supervision |

Figure 4: **Speaker identity supervision experiment.** Our Neural experts audio signal reconstruction with and without speaker identity segmentation supervision on the two speaker waveform. Colors represent expert number. The results show that the manager network is able to allocate an expert to each speaker while not compromising the reconstruction quality.

## 4.3 Surface reconstruction results

We evaluate the surface reconstruction performance of our method by fitting signed distance functions (SDFs) on four shapes from the Stanford 3D scanning repository[2]. We focus on the quality of the reconstructed surface of the shape (i.e., the zero level set of the SDF). Therefore, we follow the setup from BACON [22] and optimise points primarily around the surface of the shape.

In particular, we sample points from the surface and add Laplacian noise with two different standard deviations to give fine samples (samples close to the surface of the shape) and coarse samples (samples far from the surface of the shape). At these locations we optimise for the error between the SDF values and their ground truth SDF value. We provide further implementation details in the supplemental material.

We compare our method to a SIREN network of similar number parameters at two different sizes, called Small and Large. The small and large variants have 8 layers with 256 and 512 hidden units respectively. Our method uses 8 experts and random pretraining, with the Large size having 512 units in the encoder, 64 units in the experts and 128 units in the manager, and the Small size having 256 units in the encoder, 32 units in the experts and 64 units in the manager. As we only care about where the surface is (or equivalently, the segmentation into inside and outside) we report IoU between the inside of the predicted shape and the inside of the ground truth shape. This is calculated on a $128^3$ grid as per BACON. However, as the volume of the shapes are quite large, IoU is not a good measure of the quality of their boundary [10]. As a result, following the segmentation literature we use Trimap IoU [20, 9, 10] with different boundary distances $d$, where Trimap IoU with distance $d$ calculates IoU only on the grid points that are within distance $d$ of the surface.

We show results with Trimap IoU ($d = 0.001$) and Chamfer distance in Table 4, and a qualitative comparison in Figure 5. Our method significantly outperforms the SIREN baseline at each model size, showing that our mixture of experts is more capable at capturing fine surface detail. Interestingly, our Small model performs better than the Large SIREN baseline on most shapes. We show more results (including different values of $d$) in the supplemental material.

---

[2]https://graphics.stanford.edu/data/3Dscanrep/

| Method | Params. | Armadillo | | Dragon | | Lucy | | Thai Statue | | Mean | |
|---|---|---|---|---|---|---|---|---|---|---|---|
| | | T-IoU | $d_C$ | T-IoU | $d_C$ | T-IoU | $d_C$ | T-IoU | $d_C$ | T-IoU | $d_C$ |
| SIREN Large | 1.5M | 0.6981 | 1.4983 | 0.5517 | 2.2367 | 0.7958 | 1.2030 | 0.6191 | 5.8465 | 0.6662 | 5.4029 |
| Ours Large | 1.3M | **0.9020** | **1.4975** | **0.7500** | **2.1340** | **0.8881** | **1.1322** | **0.7318** | **5.4084** | **0.8180** | **5.0905** |
| SIREN Small | 396k | 0.5968 | 2.9165 | 0.5345 | 2.3944 | 0.6787 | 1.2583 | 0.5395 | 6.7273 | 0.5874 | 6.1553 |
| Ours Small | 323k | **0.8200** | **1.5086** | **0.6800** | **2.2102** | **0.8272** | **1.1047** | **0.5787** | **5.9721** | **0.7265** | **5.1845** |

Table 4: **3D Shape reconstruction.** Reporting the Trimap IoU for $d = 0.001$ (T-IoU↑) and Chamfer distance $\times 1e5$ ($d_C$↓) for different shapes. The results show a significant boost in performance compared to a larger MLP-based model with the same activation.

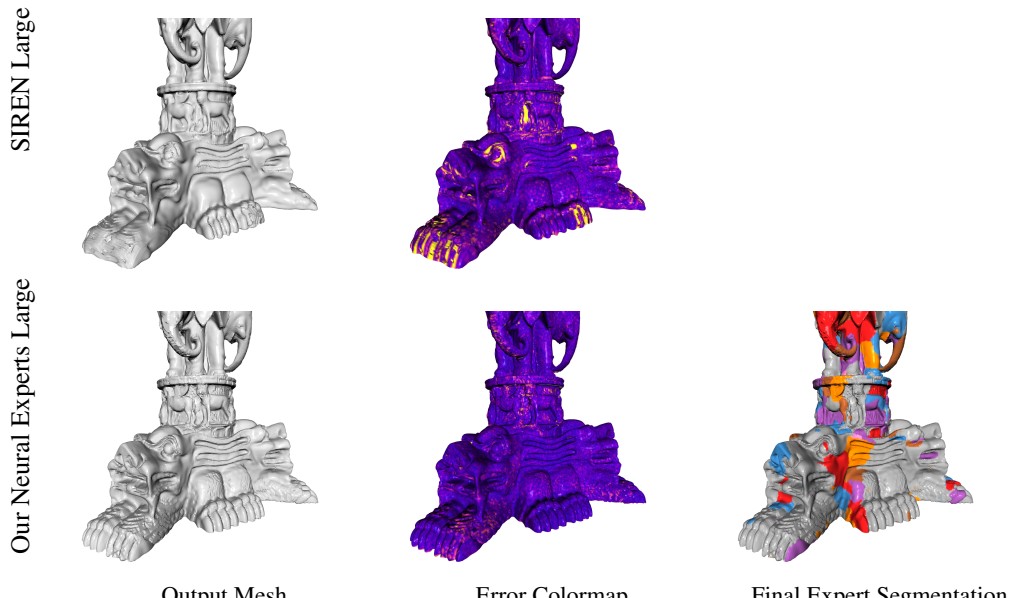

Output Mesh        Error Colormap        Final Expert Segmentation

Figure 5: **3D surface reconstruction.** Results on the Thai Statue shape. Our method noticeably captures more detail in the toes, nostrils, and eye. The error colormap shows that our method produces a mesh with far less large errors (lighter indicates higher distance to the ground truth surface), and the expert segmentation shows our method provides a subdivision of the space.

## 4.4 Ablation study

**Conditioning.** We analyze the effect of different conditioning options including no conditioning (Naive MoE), max or mean pooling over the expert encoder output and adding the output to the manager's encoder output, and concatenating the encoder outputs. The results, reported in Table 5 show that without conditioning, the method does not perform as well. Most importantly, concatenating the outputs of the encoders yields the best performance.

| Conditioning | PSNR ↑ |
|---|---|
| No conditioning | 64.52 |
| max | 77.3 |
| mean | 77.62 |
| concatenate | **81 .17** |

Table 5: **Conditioning ablation**. Without conditioning yields the worst performance however concatenation the experts encoder and manager encoder outperforms all variations.

| # Pretraining | PSNR ↑ |
|---|---|
| Fixed subdivision | 53.39 |
| Fixed random | 53.58 |
| None | 56.31 |
| SAM | 58.88 |
| Kmeans | 63.62 |
| Grid | 67.82 |
| Random | **81.17** |

Table 6: **Pretraining analysis**. Pretraining the manager to output a random expert assignment map is crucial for our method's performance.

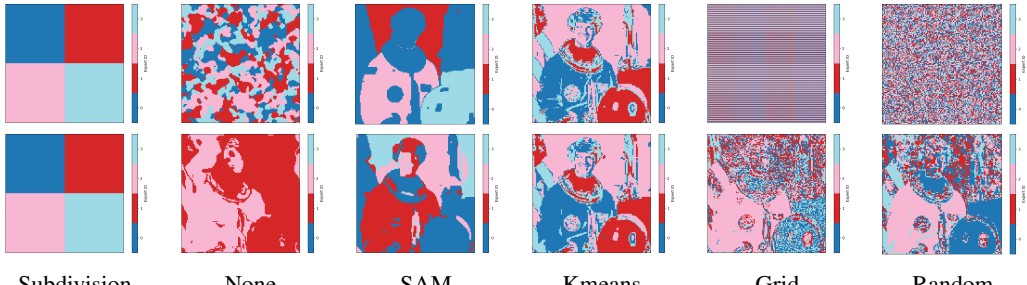

| Subdivision | None | SAM | Kmeans | Grid | Random |

Figure 6: **Manager pretraining.** Visualizing the experts selected by the manager after the pretraining stage (top row) and after the full network training (bottom row) for different pretraining ablations.

| # Experts | Params. | Time ↓ | PSNR ↑ |
|---|---|---|---|
| 2 | 266K | 18.3 | 65.57 |
| 3 | 316K | 22.3 | 71.70 |
| 4 | 366K | 26.6 | 81.17 |
| 5 | 416K | 29.7 | 77.08 |
| 6 | 466K | 33.3 | 82.68 |
| 7 | 516K | 36.5 | 82.34 |

Table 7: **Number of experts ablation**. There is a trade-off between increasing the number of experts (leading to higher parameters count and training time), and the improvement in PSNR performance. Time is iteration time (ms).

| # | Number of layers | | | |
|---|---|---|---|---|
| Experts | 1 | 2 | 4 | 8 |
| 2 | 79.39±8.74 | 77.78±8.18 | 72.23±6.92 | 65.32±6.51 |
| 4 | 92.32±7.80 | 89.35±7.10 | 81.19±8.04 | 74.36±7.31 |
| 6 | **99.15±5.32** | 92.84±7.95 | 87.80±8.98 | 75.26±9.02 |
| 8 | 96.13±7.95 | 90.72±10.6 | 83.05±10.3 | 66.57±8.72 |

Table 8: **Experts vs. Layers ablation**. The results show a trend that fewer layers and more experts yield better performance.

**Manager pretraining.** We experiment with several different manager pretraining options including none, grid, random, kmeans segmentation, and SAM segmentation [19] pretraining, presented in Figure 6. The pretraining aims to give a specific and predetermined expert assignment by the manager at initialization, so grid means that each pixel is assigned to the same expert based on a grid pattern, random uses a fixed sample from a uniform distribution for each expert, and the segmentation variants use a segmentation algorithm to determine the assignment. As an additional baseline we also report a constant manager that does not train at all and defacto acts as a constant routing component. Figure 6 also shows the manager assignment after the full training procedure. It can be seen that the manager is able to move away from its initialization and learn to cluster regions based on the input signal. The results, presented in Table 6, show that pretraining the manager is crucial for the convergence of the proposed Neural Experts and that random pretraining performs best. It seems that a balanced assignment plays an important role, as highlighted by the grid and random results. Surprisingly, semantic segmentation does not provide a significant benefit for the reconstruction task, however, may still be useful for other tasks since it provides an INR per semantic entity (the expert). Note that except for SAM and Kmeans, pretraining the manager is independent of the reconstruction signal and can be done once and used as an initialization.

**Number of Experts.** We evaluate the performance of our method given the same architecture for the encoders and manager (excluding the manager's output layer) but with different numbers of experts. The PSNR results for the astronaut image are reported in Table 7. It is important to note

| Allocation Type | Expert Encoder | Experts | Manager | Parameters | PSNR |
|---|---|---|---|---|---|
| Larger Manager | 14% | 38% | 48% | 366.1k | 83.05±10.1 |
| Larger Experts (Ours) | 14% | 55% | 31% | 366.0k | 89.35±7.10 |
| Larger Encoder | 54% | 29% | 17% | 368.3k | 88.12±6.92 |
| Balanced | 32% | 34% | 34% | 368.0k | 87.86±8.98 |

Table 9: **Parameter allocation ablation**. Neural Experts perform comparably (regardless of allocation), except in the case where the majority of the parameters are allocated to the manager.

that as more experts are introduced, the training becomes slightly noisier due to multiple pixel reassignments at every iteration. The results show a trade-off between the number of parameters and performance with more than double the training time for seven experts compared to two experts while also demonstrating an increase in PSNR of 20.55. Therefore, for our final model we opted to use four experts as a compromise between the desired performance and reasonable training time. It is important to highlight that while the number of experts has a significant impact at training time (since all experts are evaluated for training), the impact on inference time is negligible since only the selected expert is evaluated.

**Experts vs Layers.** We evaluate the performance of our method with different expert count and expert architectures (layers count and width) while keeping the number of parameters approximately constant ($\sim 366k$). The results, reported in Table 8, show a trend that less layers and more experts yield better performance for image reconstruction. Note that for our experiments we chose to use 4 experts with two layers as it provided robust performance across different modalities but for the specific case of image reconstruction 6 experts with a single layer yield the best performance.

**Parameter allocation.** In Table 9, we report results of the parameter allocation experiment, exploring the balance between the expert encoder, manager, and experts. With an approximately constant parameter count, we vary layer and element numbers to meet specific ratios. Results show robust performance across allocations, except when the manager is larger, as expected due to reduced signal capacity. See supplementary for architecture details.

## 5 Conclusion

In this paper, we introduced a novel approach to implicit neural representations (INRs) through the incorporation of a mixture of experts (MoE) architecture. Our method addresses the limitations of traditional single-network INRs by enabling the learning of local piece-wise continuous functions. This approach facilitates domain subdivision, local fitting, and opens the potential for localized editing, which is a significant advancement over existing global constraint methods.

Our results demonstrate that the Neural Experts framework significantly enhances accuracy, and memory efficiency across various reconstruction tasks, including 3D surfaces, image, and audio reconstruction. The incorporation of conditioning and pretraining methods for the manager network further improves are key elements, ensuring that the network reaches the desired solutions more effectively. Through extensive evaluations, we have shown that our MoE-INR approach outperforms traditional non-MoE methods in both performance and computational efficiency.

Future work will focus on further exploring the potential of local editing capabilities. For example, our approach can be used alongside diffusion models to provide local signal generations. Additionally, our method can be extended to more complex and larger-scale models utilizing parallelization tools, e.g., sharding. We believe that the principles and techniques introduced in this paper can pave the way for more advanced and efficient INRs, facilitating broader adoption and application in diverse fields such as computer vision, graphics, and signal processing.

**Limitations and negative impact.** Our method builds upon and extends existing INR approaches, and thus, it may inherit some limitations of the underlying networks. For example, it is known that SoftPlus MLPs have difficulty fitting low-frequency signals and therefore our SoftPlus MoE INR has the same limitation. Moreover, while increasing the number of experts in the current formulation enables parallelization and reduces computation time during inference, it also leads to slower training times as each additional expert requires more computations during training. This limitation is well known for MoE architectures and have been addressed in the literature [38, 21] for non-INR architectures and we believe this is an interesting avenue for future works.

Our proposed Neural Experts approach enhances signal reconstruction in speed and accuracy, but it also brings potential societal impacts. Efficiently encoding data into neural network formats raises concerns about digital impersonation, unauthorized data replication, and harmful content creation. Additionally, learning 3D shapes as signed distance functions (SDFs) facilitates rendering objects, which could be exploited for DeepFakes and deceptive content, posing ethical and privacy risks. While these potential misuses are speculative and require advancements beyond our current method, it is important to remain vigilant and develop safeguards to ensure responsible use of this technology.

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

# A  Appendix / supplemental material

Below we provide the architecture and training details used in the different experiments throughout the paper as well as additional experimental results and visualizations.

## A.1  Image reconstruction

**Implementation details.** In Section 4.1 we reported the performance of the proposed approach compared to other prominent architectures. For our approach we used 4 experts, 2 hidden layers for encoder and 2 for the experts. The manager has a similar architecture with 2 layers for the manager encoder and 2 for the final manager block. Each layer has 128 elements. All models were trained using an Adam optimizer, a learning rate of $10^{-5}$ with exponential decay. All models are trained for 30K iterations where for our approach we use $t_{all} = 80\%$ and $t_e = 20\%$, i.e. we train all parameters for the first 24K iteration and just the experts for the remaining 6K iterations. These models were trained on an NVIDIA A5000 GPU. All input images to the INR were cropped to be in a 1:1 ratio, scaled down by a factor of four and their coordinates were scaled to be in the unit cube to accelerate training (following [22]).

For a fair comparison we run a vanilla MLP with Sine or Softplus activations with a total of 4 hidden layers and 128 elements in each layer. However, this yields significantly fewer parameters so we implement a 'Wider' version that will have a similar 'width' as all experts combined width. Therefore the 'Wider' version has 512 elements for the last two hidden layers. This yields improved performance compared to the vanilla (naive) MoE implementation. However, when compared to our Neural Experts, even with its higher parameter count 'Wider' underperforms.

**Additional results.** We report the performance of various activation functions and approaches on image reconstruction in Table 10. We extend the results in the main paper and include results with InstantNGP as an example of newer hybrid INRs that use parameters for the encoding. The results show that our Neural Experts SIREN and Neural Experts FINER achieve superior performance compared to InstantNGP (89.35 and 90.84 PSNR vs 84.56 PSNR) with an order of magnitude fewer parameters (366k vs 7.7M). In the InstantNGP experiment, we use their default architecture for image experiments (16 encoding levels, 2 parameters per level, maximum cache size of $2^{19}$, and a 2 hidden layer decoder with 64 neurons). Note that most of InstantNGP's parameters are spatial parameters (hash encoding), specifically 99.86% of all parameters. This comparison between our Neural Experts SIREN/FINER and InstantNGP shows that the spatial parameters (parameters dedicated for specific spatial regions, so the experts and the hash encoding) are effective, but if used with patterns and heuristics, many of the spatial parameters become redundant and underutilized quickly. This motivates the learning of the spatial regions to achieve a parameter efficient approach (our Neural Experts method).

Extending InstantNGP to include the MoE architecture is non-trivial as the hash encoding parameters, which make up over 99% of the total parameters, are used in the beginning of the network (to provide a specialized encoding to then go through a tiny MLP decoder). The obvious Neural Experts extension (which requires a shared input encoding), is to apply MoE to that tiny decoder, which we call Naive Neural Experts InstantNGP. We observe that the baseline InstantNGP performs better than this variant. However, the shared input hash encoding parameters are still over 99% of the total parameters. In the parameter allocation experiment Table 11, the allocation ratio between the expert encoder, the experts and the manager plays an important role in the MoE's performance, with the best performing ratio being 14%, 55% and 31% respectively. So this naive result is to be expected. Constructing a more fair distribution of parameters would require a fundamental change to the InstantNGP backbone. Either we greatly increase the parameters of the expert and manager (which would make the total parameters another order of magnitude larger) or we need a way to split the hash encoding parameters into the experts. Both of these extensions are outside the scope of the current work and we leave this for future work.

In Table 11 we extend Table 9 to provide exact architecture details and include the sizes of each component: expert encoding, experts, manager encoding and manager. These are given in the form of width x layers except for experts which is given as number of experts x ( width x layers ).

Here, we also provide the results of our method on additional images from the Kodak dataset [12]. The results show that our method achieves improved performance consistently.

| Activation | Method | Params. | PSNR Mean | Std | SSIM Mean | Std | LPIPS Mean | Std |
|---|---|---|---|---|---|---|---|---|
| Softplus | Base | 99.8k | 19.51 | 2.95 | 0.7158 | 0.1106 | 4.08e-1 | 8.33e-2 |
| Softplus | Wider | 642.8k | 20.91 | 3.12 | 0.7798 | 0.0899 | 3.38e-1 | 8.44e-2 |
| SoftPlus | Ours | 366.0k | 20.62 | 3.12 | 0.7628 | 0.0946 | 3.53e-1 | 8.47e-2 |
| Softplus + FF | Base | 99.8k | 28.97 | 3.30 | 0.9433 | 0.0193 | 7.59e-2 | 1.70e-2 |
| Softplus + FF | Wider | 642.8k | 29.48 | 3.91 | 0.9436 | 0.0245 | 7.74e-2 | 2.30e-2 |
| SoftPlus + FF | Ours | 366.0k | 31.66 | 3.16 | 0.9652 | 0.0180 | 4.13e-2 | 1.57e-2 |
| Sine | Base | 99.8k | 57.23 | 2.46 | 0.9991 | 0.0005 | 5.78e-4 | 5.09e-4 |
| Sine | Wider | 642.8k | 77.50 | 5.32 | 0.9996 | 0.0005 | 3.08e-4 | 3.04e-4 |
| Sine | V. MoE | 349.6k | 62.98 | 4.16 | 0.9993 | 0.0005 | 4.53e-4 | 3.88e-4 |
| Sine | V. MoE + RP | 349.6k | 74.28 | 7.36 | 0.9996 | 0.0004 | 3.05e-4 | 2.88e-4 |
| Sine | Ours small | 98.7k | 63.42 | 7.09 | 0.9992 | 0.0007 | 9.96e-4 | 2.40e-3 |
| Sine | Ours | 366.0k | 89.35 | 7.10 | **0.9997** | 0.0004 | 2.49e-4 | 2.93e-4 |
| FINER | Base | 99.8k | 58.08 | 3.04 | 0.9991 | 0.0005 | 7.47e-4 | 1.31e-3 |
| FINER | Wider | 642.8k | 80.32 | 5.40 | 0.9996 | 0.0004 | 2.49e-4 | 2.46e-4 |
| FINER | Ours | 366.0k | **90.84** | 8.14 | **0.9997** | 0.0004 | **2.46e-4** | 2.48e-4 |
| InstantNGP | Base | 7.7M | 84.56 | 5.62 | 0.9996 | 0.0003 | 4.28e-4 | 5.37e-4 |
| InstantNGP | Naive NE | 7.7M | 75.14 | 2.57 | 0.9996 | 0.0004 | 4.07e-4 | 4.35e-4 |

Table 10: **Image reconstruction.** Comparing all approaches on the Kodak dataset [12]. Our approach reconstructs the image more faithfully and outperforms the baselines. FF: Fourier Features, RP: Random Pretraining, V. MoE: Vanilla MoE.

| Allocation Type | Architecture Exp. Enc., Exp., Man. Enc., Man. | Percentages Exp. Enc. | Exp. | Man. | Parameters | PSNR |
|---|---|---|---|---|---|---|
| Larger Manager | 128x2, 4x(102x2), 128x2, 196x2 | 14% | 38% | 48% | 366.1k | 83.05±10.11 |
| Larger Experts (Ours) | 128x2, 4x(128x2), 128x2, 128x2 | 14% | 55% | 31% | 366.0k | 89.35±7.10 |
| Larger Encoder | 256x2, 4x(68x2), 96x2, 68x2 | 54% | 29% | 17% | 368.3k | 88.12±6.92 |
| Balanced | 196x2, 4x(85x2), 128x2, 128x2 | 32% | 34% | 34% | 368.0k | 87.86±8.98 |

Table 11: **Parameter allocation ablation**. We update Table 9 to include the sizes for each component.

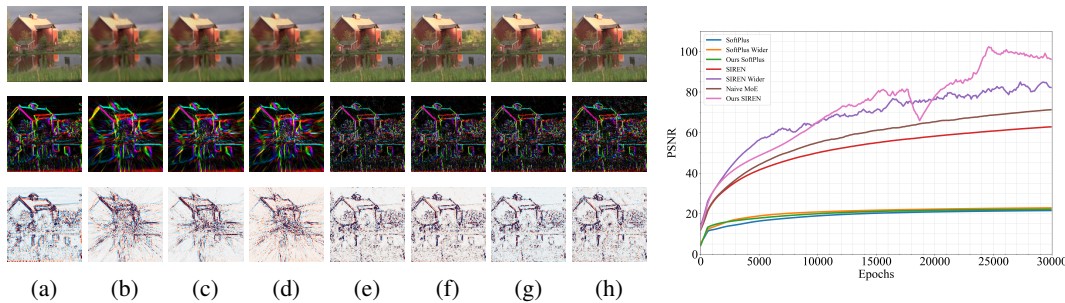

(a) (b) (c) (d) (e) (f) (g) (h)

Figure 7: Image reconstruction qualitative (left) and quantitative (right) results. Showing image reconstruction (top), gradients (middle), and laplacian (bottom) for (a) GT, (b) SoftPlus, (c) SoftPlus Wider, (d) Our SoftPlus MoE, (e) SIREN, (f) SIREN Wider, (g) Naive MoE, and (h) Our SIREN MoE. The quantitative results (right) report PSNR as training progresses and show that our MoE architecture outperforms all baselines.

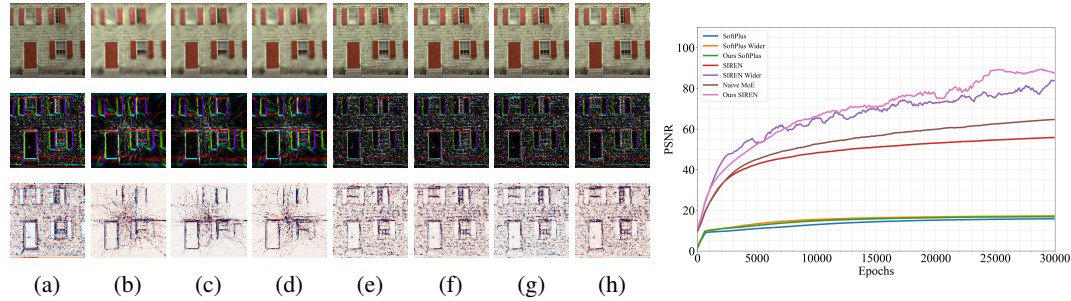

(a) (b) (c) (d) (e) (f) (g) (h)

Figure 8: Image reconstruction qualitative (left) and quantitative (right) results. Showing image reconstruction (top), gradients (middle), and laplacian (bottom) for (a) GT, (b) SoftPlus, (c) SoftPlus Wider, (d) Our SoftPlus MoE, (e) SIREN, (f) SIREN Wider, (g) Naive MoE, and (h) Our SIREN MoE. The quantitative results (right) report PSNR as training progresses and show that our MoE architecture outperforms all baselines.

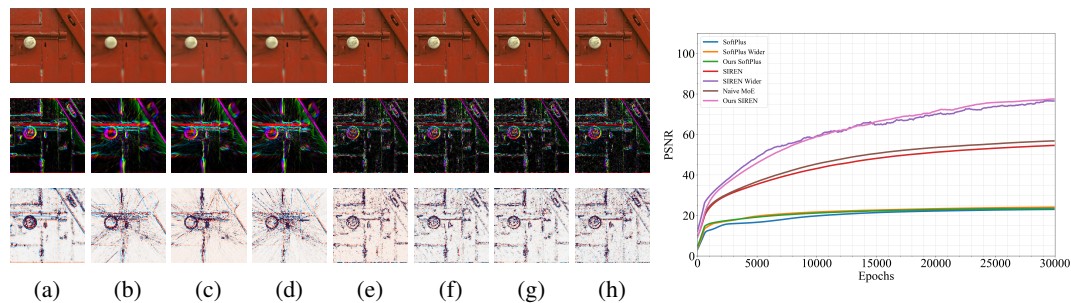

(a) (b) (c) (d) (e) (f) (g) (h)

Figure 9: Image reconstruction qualitative (left) and quantitative (right) results. Showing image reconstruction (top), gradients (middle), and laplacian (bottom) for (a) GT, (b) SoftPlus, (c) SoftPlus Wider, (d) Our SoftPlus MoE, (e) SIREN, (f) SIREN Wider, (g) Naive MoE, and (h) Our SIREN MoE. The quantitative results (right) report PSNR as training progresses and show that our MoE architecture outperforms all baselines.

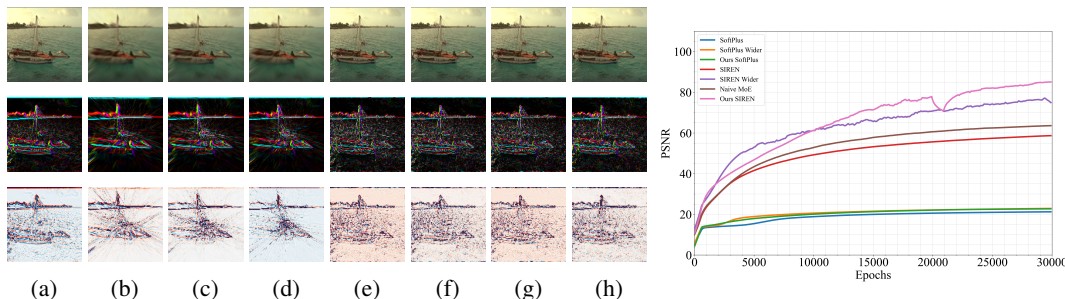

(a) (b) (c) (d) (e) (f) (g) (h)

Figure 10: Image reconstruction qualitative (left) and quantitative (right) results. Showing image reconstruction (top), gradients (middle), and laplacian (bottom) for (a) GT, (b) SoftPlus, (c) SoftPlus Wider, (d) Our SoftPlus MoE, (e) SIREN, (f) SIREN Wider, (g) Naive MoE, and (h) Our SIREN MoE. The quantitative results (right) report PSNR as training progresses and show that our MoE architecture outperforms all baselines.

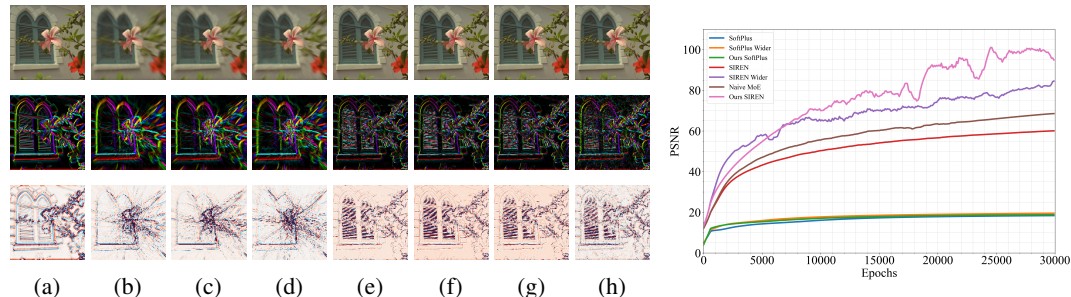

|     |     |     |     |     |     |     |     |
| --- | --- | --- | --- | --- | --- | --- | --- |
| (a) | (b) | (c) | (d) | (e) | (f) | (g) | (h) |

Figure 11: Image reconstruction qualitative (left) and quantitative (right) results. Showing image reconstruction (top), gradients (middle), and laplacian (bottom) for (a) GT, (b) SoftPlus, (c) SoftPlus Wider, (d) Our SoftPlus MoE, (e) SIREN, (f) SIREN Wider, (g) Naive MoE, and (h) Our SIREN MoE. The quantitative results (right) report PSNR as training progresses and show that our MoE architecture outperforms all baselines.

## A.2 Audio signal reconstruction experiments

Here, we provide qualitative results of our method compared to prominent baselines in Figure 12. The results show that our proposed approach provides reduced errors. We also provide the resulting audio reconstruction `.wav` files in our supplementary material `.zip` file.

Note that the architectural and training details for the audio signal reconstruction experiment are the same as in the image reconstruction experiments.

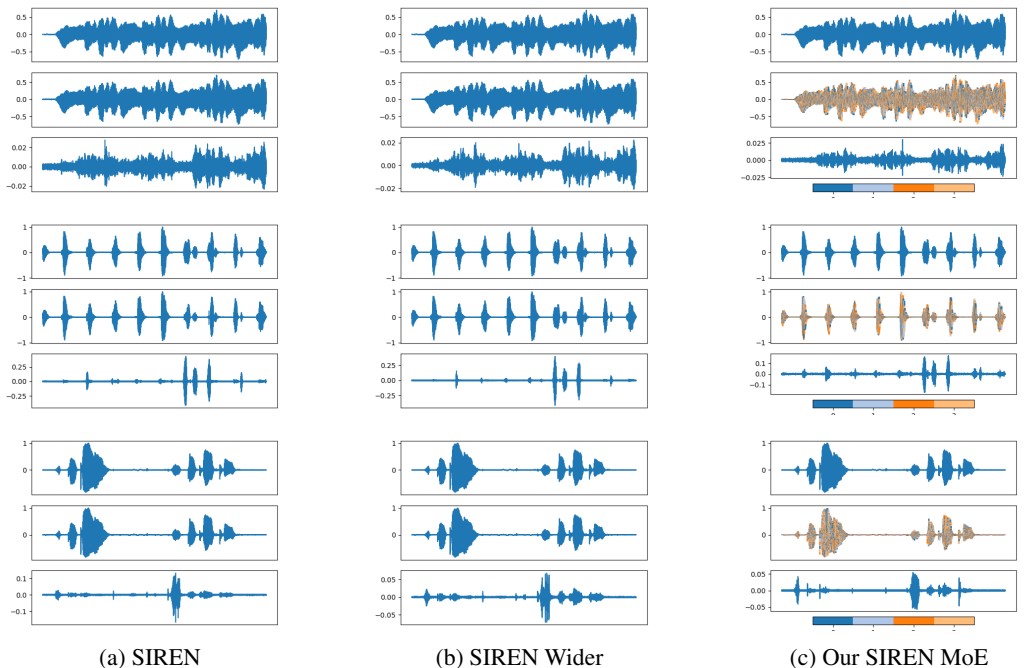

          (a) SIREN                 (b) SIREN Wider            (c) Our SIREN MoE

Figure 12: **Audio reconstruction visualization.** Bach, counting, and two spreakers audio are presented in the top, middle and bottom row respectively. Within each waveform block, the rows represent the ground truth, reconstruction, and error visualization from top to bottom. For our Neural Experts we color code the different experts on the reconstructed waveform.

|  | | Mean | | | |
| Method | # params | IoU | IoU(0.1) | IoU(0.01) | IoU(0.001) |
|---|---|---|---|---|---|
| SIREN Large | 1.5M | 0.9865 | 0.9886 | 0.9576 | 0.6662 |
| MoE Large | **1.3M** | **0.9912** | **0.9936** | **0.9770** | **0.8180** |
| SIREN Small | 396k | 0.9786 | 0.9802 | 0.9235 | 0.5874 |
| MoE Small | **323k** | **0.9834** | **0.9853** | **0.9476** | **0.7265** |

Table 12: IoU and Trimap IoUs of 3D reconstruction. IoU($d$) indicates Trimap IoU with boundary region of radius $d$ (IoU is computed within the region of distance $d$ of the ground truth boundary).

|  | | Armadillo | | Dragon | | Lucy | | Thai Statue | | Mean | |
| Method | # params | IoU | IoU(0.001) | IoU | IoU(0.001) | IoU | IoU(0.001) | IoU | IoU(0.001) | IoU | IoU(0.001) |
|---|---|---|---|---|---|---|---|---|---|---|---|
| SIREN Large | 1.5M | 0.9963 | 0.6981 | 0.9861 | 0.5517 | 0.9930 | 0.7958 | 0.9706 | 0.6191 | 0.9865 | 0.6662 |
| MoE Large | **1.3M** | **0.9982** | **0.9020** | **0.9915** | **0.7500** | **0.9970** | **0.8881** | **0.9782** | **0.7318** | **0.9912** | **0.8180** |
| SIREN Small | 396k | 0.9920 | 0.5968 | 0.9785 | 0.5345 | 0.9855 | 0.6787 | **0.9583** | 0.5395 | 0.9786 | 0.5874 |
| MoE Small | **323k** | **0.9977** | **0.8200** | **0.9899** | **0.6800** | **0.9950** | **0.8272** | 0.9509 | **0.5787** | **0.9834** | **0.7265** |

Table 13: IoU and Trimap IoU of 3D reconstruction. IoU($d$) indicates Trimap IoU with boundary region of radius $d$ (IoU is computed within the region of distance $d$ of the ground truth boundary).

## A.3 Surface Reconstruction

**Implementation Details.** We first scale the input shape points to fit within the $[-1, 1]^3$ cube as this is the recommended domain for SIREN [39], and take the domain to be the $[-1.2, 1.2]^3$ cube to ensure spacing around the scaled points. As this is larger than the $[-0.5, 0.5]^3$ cube used in BACON [22], we double the standard deviation they use for the coarse and fine samples ($4 \cdot 10^{-2}$ and $4 \cdot 10^{-6}$ respectively). We use an L1 loss and weight all points equally rather than an L2 loss and weight closer points more as done in BACON, as we find the former works better (for both the baseline and our method). We use 10k surface samples, 10k fine samples and 10k coarse samples per batch. We optimize our network for 30000 iterations using Adam, starting from a learning rate of $5 \cdot 10^{-3}$ and decreasing the learning rate by 0.9999 at each iteration. We run our surface reconstruction experiments on a single RTX 3090 (24GB VRAM).

In BACON, the ground truth SDF for a point is approximated by finding the three closest points in the input and averaging their normal to determine the sign (whether the initial point is inside or outside the shape), while taking the distance to the closest point as the unsigned distance. We find that this causes spurious inside regions along medial axes where the mean of the normals cancel out, and instead use the closest point.

The random pattern in the manager pretraining is done by dividing the domain into a $64^3$ grid and assigning random experts for each grid cell. The IoU is computed on a $128^3$ grid of the domain as per BACON, taking the grid points that predicted as inside as the set of interest. Trimap IoU with distance $d$ is computed by only using grid points whose distance to the surface is less than or equal to $d$ (as determined by the ground truth SDF at those grid points).

Visualization is done by evaluating on a $512^3$ grid and running marching cubes [25].

**Further Results and Visualizations.** We report IoU and Trimap IoU with different distance $d$ in Table 12 and Table 13. We also provide visualizations for each shape in Figure 13, Figure 14, Figure 15 and Figure 16.

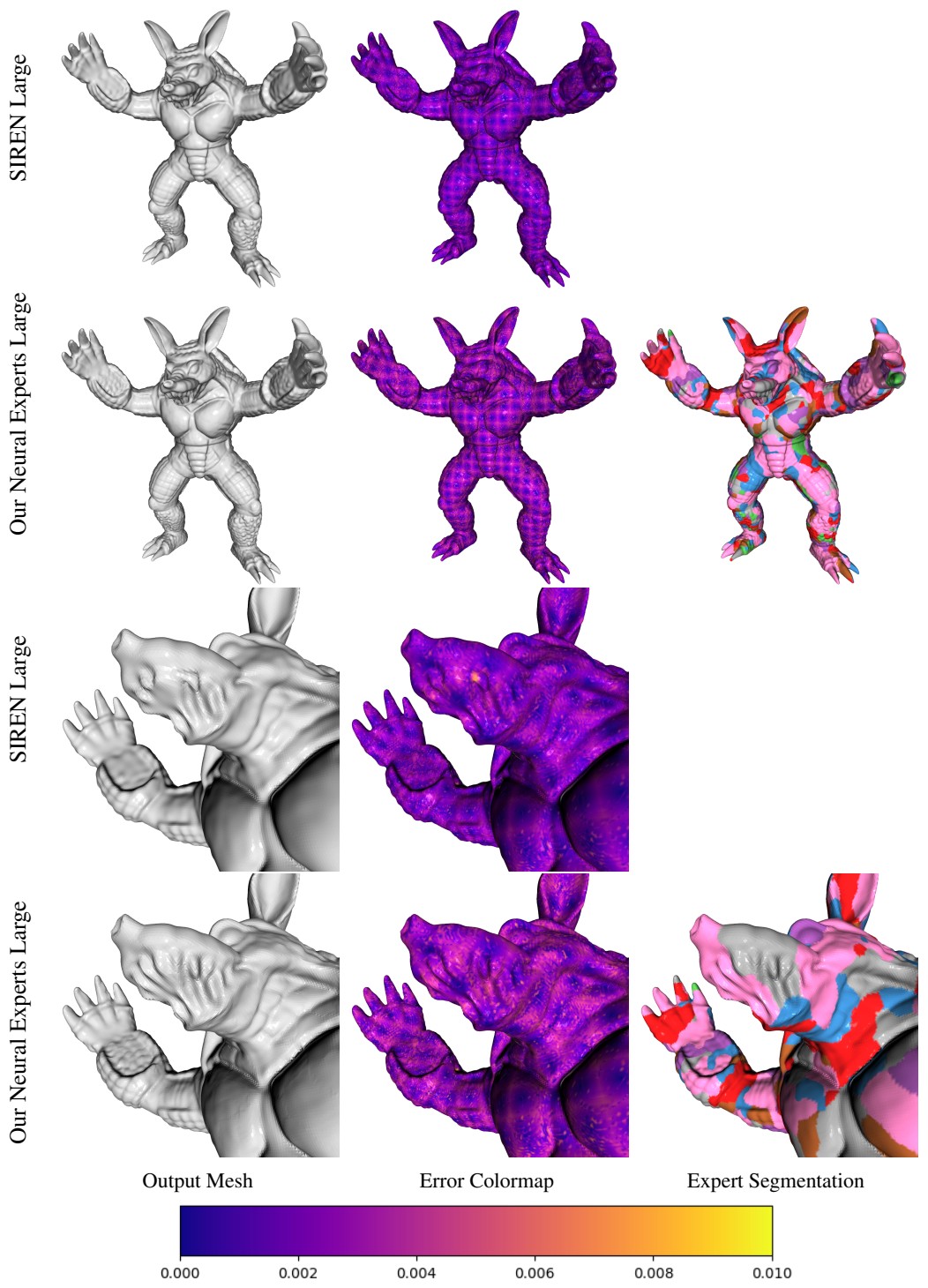

Figure 13: Results on the Armadillo shape. Our method performs better around the mouth and forearm region where there are high levels of detail. SIREN Large fits an oversmooth surface plus small inside surfaces in order to capture indentations, leading to poor Trimap IoU performance.

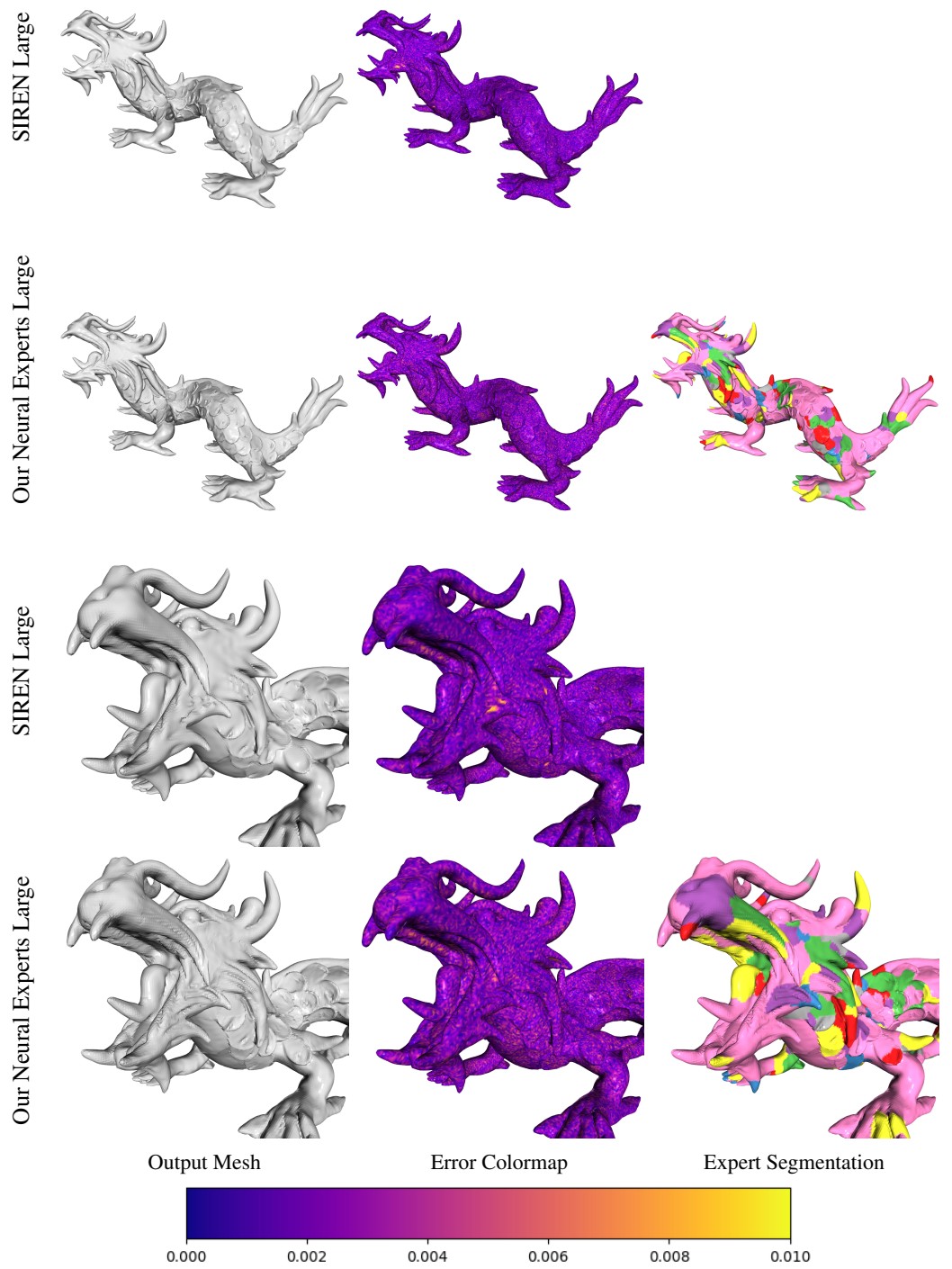

Figure 14: Results on the Dragon shape. Our method captures more detail around the jaws such as indentations.

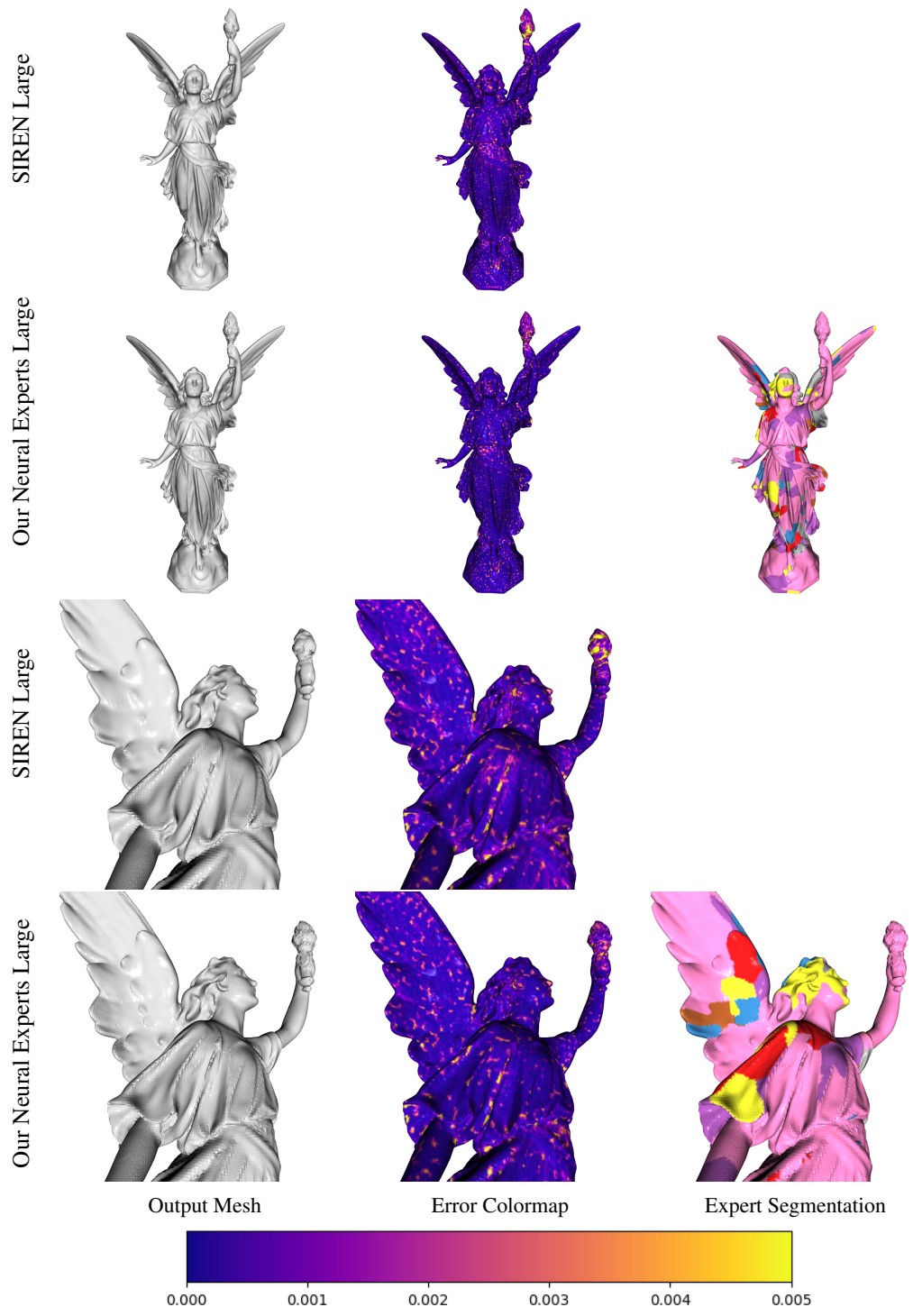

Figure 15: Surface reconstruction results on the Lucy shape. Our method performs overall better but in particular at the torch region where there are high levels of detail.

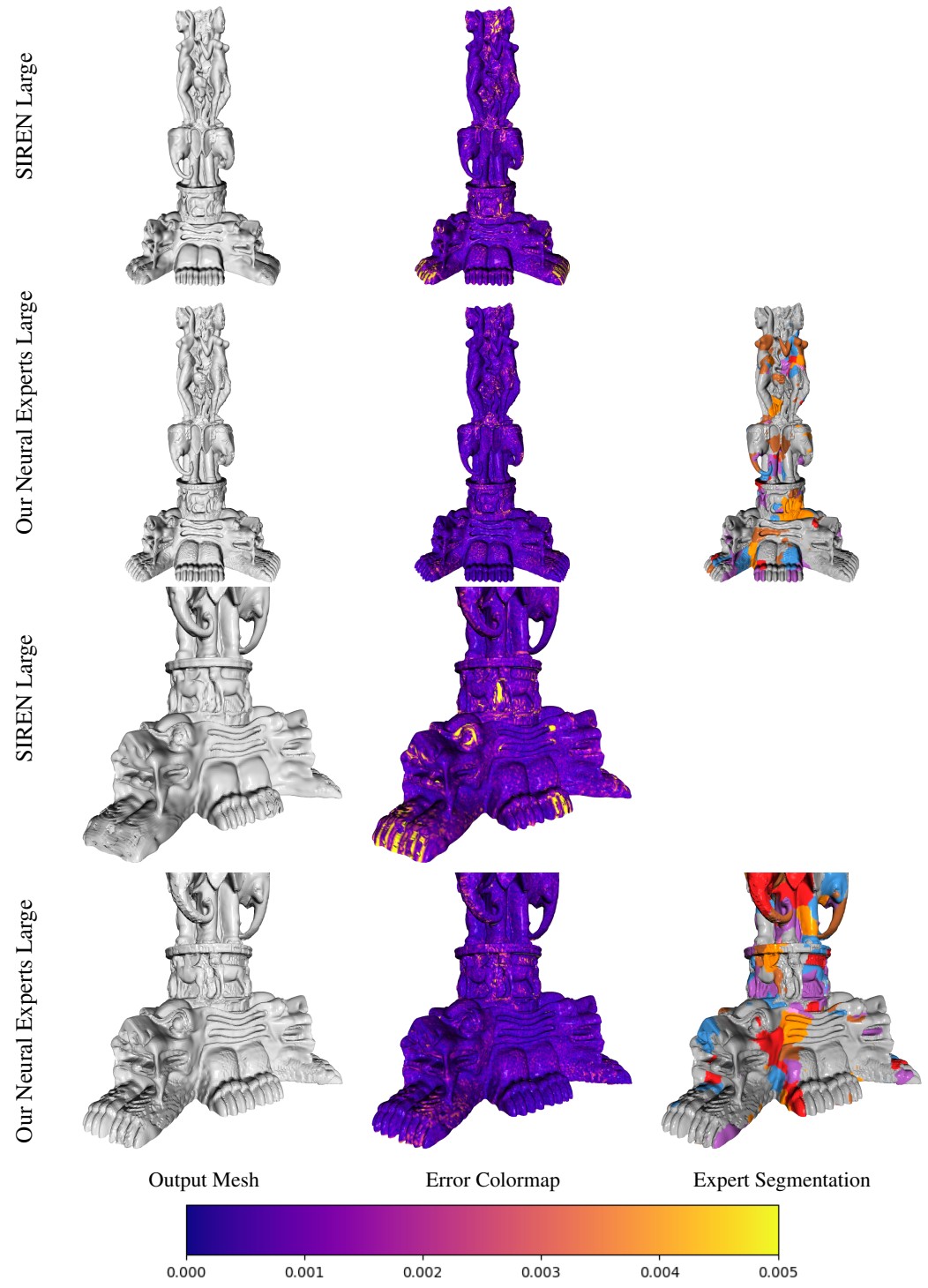

Output Mesh       Error Colormap       Expert Segmentation

0.000    0.001    0.002    0.003    0.004    0.005

Figure 16: Results on the Thai Statue shape. Our method noticeably captures more detail in the toes, nostrils, and eye.

