# OpenReview forum: "Neural Experts: Mixture of Experts for Implicit Neural Representations"
_NeurIPS.cc/2024/Conference — NeurIPS 2024 poster_

### Official Review · Reviewer_ekS9 · 2024-06-27

**Soundness:** 2
**Presentation:** 3
**Contribution:** 2
**Rating:** 5
**Confidence:** 5

**Summary:**

This paper proposes a mixture of experts (MoE) approach for INRs, which allows the learning of local piece-wise continuous functions by subdividing the domain and fitting locally. The incorporation of a MoE architecture enhances speed, accuracy, and memory efficiency. They also propose a novel manager architecture and initialization that enable domain subdivision without ground truth.

**Strengths:**

1. The paper is well-written and easy to follow.
2. The proposed MoE INR has a good performance compared to baselines.
3. The idea of delivering MoE as a learnable partition region for INR fitting with randomized initialization is novel to me. From the ablation study, the randomized initialization improves the performance a lot.

**Weaknesses:**

1. Missing some closely-relative works. I encourage the authors to have a detailed discussion of previous MoE INRs [1,2] and decomposition/partition-based INRs [3,4].
2. Lacking some key comparison experiments with decomposition/partition-based INRs. The authors only compare their method with the baseline SoftPlus and SIREN (and their wider version). However, some related works [4] have also shown the INR based on pre-determined masks can also outperform the wider version of SIREN.  I encourage the authors to experimentally compare your method with [4] to illustrate the necessity of learnable partition regions.
3. A detailed ablation study on the hyper-parameters of the MoE INRs is missing, such as the layer of encoder, manager, and experts. Given a fixed number of parameters, how to allocate the parameters to the three modules remains unknown.

[1] Zhao, Jianchen, et al. "MoEC: Mixture of Experts Implicit Neural Compression." arXiv preprint arXiv:2312.01361 (2023).
[2] Wang, Peihao, et al. "Neural implicit dictionary learning via mixture-of-expert training." International Conference on Machine Learning. PMLR, 2022.
[3] Rebain, Daniel, et al. "Derf: Decomposed radiance fields." Proceedings of the IEEE/CVF Conference on Computer Vision and Pattern Recognition. 2021.
[4] Liu, Ke, et al. "Partition speeds up learning implicit neural representations based on exponential-increase hypothesis." Proceedings of the IEEE/CVF International Conference on Computer Vision. 2023.

**Questions:**

1. Could you please discuss how to choose the number of experts giving a fixed number of parameters and iteration time? Is it better to have more experts with each one having fewer parameters or fewer experts with each one having more parameters?
2. I wonder whether it is possible to apply your method to NeRF since there are no supervised signals for the 3D ground truth.
3. When comparing the Vanilla MoE INR and your Neural Experts, have you kept their total parameters similar (smaller experts for your Neural Experts due to the extra parameters needed by the encoders)?

**Limitations:**

The limitations and potential negative societal impact of their work have been discussed in the paper.

---

> ### Author Rebuttal · Authors · 2024-08-07
>
> W1: Thank you for bringing these important works to our attention, we will address them in our paper.
>
> W2: We compare with [4,new] on their dataset (300 test images from LSUN bedrooms). We use the same learning rate and same measure (PSNR after 300 steps for SIREN). Note that these have not converged at 300 iterations (see also Fig. 5 in [4,new]), hence for the experiments in our paper we report after 30k steps (also done for the last row in the table).
>
> | **Method**                        | **Mean** | **Std** | **Parameters** | **Steps** |
> |-----------------------------------|----------|---------|----------------|-----------|
> | SIREN [4,new]                         | 21.211   | -       | 790.0k         | 300       |
> | SIREN-PoG [4,new]                     | 23.864   | -       | 793.6k         | 300       |
> | SIREN-PoS [4,new]                     | 24.485   | -       | 793.6k         | 300       |
> | Ours Neural Experts SIREN         | 30.50    | 2.73    | 366.0k         | 300       |
> | Ours Neural Experts SIREN         | 54.98    | 2.56    | 366.0k         | 30000     |
>
> W3:
> While keeping the number of overall parameters fairly constant, we experiment with how to allocate parameters between the components. The first column is
> `width x layers for the input encoding, num experts x (width x layers) for the experts themselves, width x layers for the manager encoding, width x layers for the manager` and the second column is
> `% parameters for input encoding, % parameters for the experts, % parameters for the manager`.
> We can see that as long as the majority of the parameters is not for the manager (one third of the total params or less) our neural experts SIREN model performs similarly.
>
>
> | **Layers**                        | **Pecentages** |**Parameters**  | **PSNR**        |
> |-----------------------------------|----------------|----------------|-----------------|
> | 128x2, 4x(128x2), 128x2, 128x2    | 14%, 55%, 31%  | 366.0k         | 89.35±7.10      |
> | 196x2, 4x(85x2), 128x2, 128x2     | 32%, 34%, 34%  | 368.0k         | 87.86±8.98      |
> | 128x2, 4x(102x2), 128x2, 196x2    | 14%, 38%, 48%  | 366.1k         | 83.05±10.11     |
> | 256x2, 4x(68x2), 96x2, 68x2       | 54%, 29%, 17%  | 368.3k         | 88.12±6.92      |
>
> Q1: We now experiment with changing the number of experts vs number of layers within the experts. Like the previous table, we keep the parameters as similar as possible (~366k) by changing the layer width of experts. The results show that less layers and somewhat high number of layers is best, with 6 experts and 1 layer performing the best out of all the configurations we tried. Our chosen configuration of 4 experts and 2 layers is somewhat consistent with this rule of thumb.
>
> | Number of experts \ Number of layers |     1       |      2       |    4        |     8      |
> |--------------------------------------|-------------|--------------|-------------|------------|
> | 2                                    | 79.39±8.74  | 77.78±8.18   | 72.23±6.92  | 65.32±6.51 |
> | 4                                    | 92.32±7.80  | 89.35±7.10   | 81.19±8.04  | 74.36±7.31 |
> | 6                                    | 99.15±5.32  | 92.84±7.95   | 87.80±8.98  | 75.26±9.02 |
> | 8                                    | 96.13±7.95  | 90.72±10.61  | 83.05±10.31 | 66.57±8.72 |
>
>
>
> Q2: It is not possible to use our method for NeRFs, for the reason the reviewer points out, see our discussion in the General Comments section above.
>
> Q3: The parameters are similar 349584 for vanilla SIREN MOE vs 365968 for our neural experts SIREN (so 95.5\% of the params of our model, or our model is a 1.05x increase in parameters).

---

> ### Comment · Reviewer_ekS9 · 2024-08-12
> **Official Comment by Reviewer ekS9**
>
> I appreciate the authors' effort in providing such a detailed rebuttal.
>
> One of my major concerns is the comparison of your paper with "MoEC: Mixture of Experts Implicit Neural Compression". It seems that their model is the vanilla MOE in your paper, right? Furthermore, I find that Random Pretraining for manager is very important to performance. When you conduct the vanilla MOE experiment, have you also applied the Random Pretraining for manager strategy? If not, I wonder whether the Random Pretraining for manager strategy can improve the performance of vanilla MOE.

---

> > ### Author Response · Authors · 2024-08-13
> >
> > **Results with Random Pretraining the Vanilla SIREN MOE and discussion of MoEC**
> > We did not apply the random pretraining method for the Vanilla SIREN MOE as it is a contribution of our method that we proposed. We give results when adding the random pretraining to the Vanilla SIREN MOE in the updated Table 1 (put as a new general comment above). We see a significant increase in performance when adding random pretraining to the Vanilla SIREN MoE (62.98 vs 74.28 PSNR) while still having lower performance than our full model (89.35 PSNR). This shows that both random pretraining and the shared input encoder are important parts of the method.
> >
> > While the MoEC architecture is similar to our Vanilla SIREN MoE baseline, it does not follow the pure MoE framework from [16,18,28] that our method uses. Instead they use the "sparsely-gated mixture of experts layer" from [35] ([28] in their paper) where conditional computation is done within the network to produce a single output (note the shared decoder in their method, compared to the expert decoders in ours). Thus they cannot use the pure MoE loss, so similar to [35] they use a balancing loss to ensure load balancing and sparsity. Furthermore they only show the efficacy of their method on a single domain (one biomedical dataset) and one base architecture, while we show the efficacy of our method on multiple domains and multiple base architectures. However their motivation is similar to ours: show that using spatial parameters with learnable partitions can improve performance over using them with fixed partitions when constraining the number of parameters.

---

### Official Review · Reviewer_ij4R · 2024-07-09

**Soundness:** 1
**Presentation:** 3
**Contribution:** 1
**Rating:** 5
**Confidence:** 4

**Summary:**

This paper proposes a new architecture for implicit neural representations (INRs) based on the mixture-of-experts (MoE) architecture. This new architecture is differs from traditional MoE architectures in that now all the experts have a shared encoder and expert-specific decoders, while the manager also now has an encoder-decoder architecture, with the manager decoder taking as input both the manager encoder's representation as well as the experts' shared encoder representation. The authors also provide a pre-training method for the manager. The method is evaluated on image reconstruction, audio reconstruction, and 3D shape reconstruction by fitting signed distance functions (SDFs).

**Strengths:**

This paper proposes a novel MoE-based architecture for INRs together with a novel pre-training strategy for the MoE manager.

The empirical results are good and there is an ablation study on the major components.

The paper is well-written and easy to understand.

Many details relating to reproducibility are provided in the supplemental material.

**Weaknesses:**

One of the major weaknesses of this paper is the experimental evaluation. The method does not compare against other methods that propose a new INR architecture (e.g. Gaussian activation function [1], WIRE [2], FINER [3]) or a standard MLP with positional encoding. The experimental evaluation is not also very robust, as only small datasets are used (Kodak, only 3 audio recordings, only 4 shapes) and more complicated tasks investigated by similar works are not considered (for example, WIRE [2] and FINER [3] both include evaluation on neural radiance fields).

The proposed MoE architecture also did not show improvements when used with softplus activation.

This paper also did not include other metrics normally used to evaluate the tasks, such as SSIM and LPIPS for 2D image fitting (e.g. FINER [3]).

Minor point: inclulding the number of parameters in Table 1 may be helpful since comparisons between the number of parameters is discussed in the text.

References
1. Ramasinghe, Sameera, and Simon Lucey. "Beyond periodicity: Towards a unifying framework for activations in coordinate-mlps." European Conference on Computer Vision. Cham: Springer Nature Switzerland, 2022.
2. Saragadam, Vishwanath, et al. "Wire: Wavelet implicit neural representations." Proceedings of the IEEE/CVF Conference on Computer Vision and Pattern Recognition. 2023.
3. Liu, Zhen, et al. "FINER: Flexible spectral-bias tuning in Implicit NEural Representation by Variable-periodic Activation Functions." Proceedings of the IEEE/CVF Conference on Computer Vision and Pattern Recognition. 2024.

**Questions:**

1. Can this work be combined with traditional ReLU MLPs with positional encoding or MLPs with activation functions other than sine?
2. How does this work compare to other novel INR architectures (e.g. WIRE, FINER, etc)?

**Limitations:**

Limitations and negative impact are addressed by the authors.

---

> ### Author Rebuttal · Authors · 2024-08-07
>
> **One of the major weaknesses of this paper is the experimental evaluation ...**
> - **vs other architectures**: All the listed architectures are about changing the activation function. Our proposal is orthogonal to that as it can work with any activation function, and we have shown our method with two activation functions. We have added two more, Softplus+FF and FINER, in the updated Table 1 (see General Comments). We also explain why we cannot test our method on NeRFs in the General Comments section above.
> - **Dataset Size**: Our experiment dataset sizes are similar to the papers you listed. In each of those, relevant experiments (Image,SDF,Audio) are done on similar or smaller sizes (only FINER's SDF experiments has more: 5 shapes to our 4 shapes). In fact many experiments and comparisons are done on as small as 1 instance. (FINER: 16 2D images and 5 SDF shapes, Gaussian activations: comparisons on 1-3 images per property, WIRE: 1 image, 1 3D shape, comparisons on 1 image per property). We compare on 24 images, 4 SDF shapes, and 3 audio samples.
>
> **The proposed MoE architecture also did not show improvements when used with softplus activation.**
> Softplus activations alone is not expressive enough. We give new results with Softplus+FF (Fourier Features) in the updated Table 1 (see general comment).
>
> **This paper also did not include other metrics normally used to evaluate the tasks, such as SSIM and LPIPS for 2D image fitting (e.g. FINER [3]).**
> - We have added these metric in the updated Table 1 (see General Comments)
>
> **Q1. Can this work be combined with traditional ReLU MLPs with positional encoding or MLPs with activation functions other than sine?**
> Yes, we give new results with Softplus+FF (Fourier Features) in the updated Table 1 (see general comment).
>
> **Q2. How does this work compare to other novel INR architectures (e.g. WIRE, FINER, etc)?**
> As mentioned before, those are orthogonal contributions as those methods change the activation function, and we can use any activation function in our framework. See our new results with FINER in the updated Table 1.

---

### Official Review · Reviewer_VtKc · 2024-07-12

**Soundness:** 2
**Presentation:** 2
**Contribution:** 3
**Rating:** 6
**Confidence:** 4

**Summary:**

The paper presents a novel INR framework that leverages the Mixture of Experts (MoE) technique. The proposed strategy consists of an expert and a manager branch. Each branch has an encoder that processes the input coordinate and extracts an embedding. By processing the two encoder embeddings, the manager predicts the probability of which of the N experts should be used for extracting the signal. They show how the proposed INR framework achieves better reconstructions than SIREN on several modalities, such as audio, image, or 3D surfaces. They also propose a manager pre-training strategy, which is necessary to exploit all the experts effectively.

**Strengths:**

-	The original idea might be conducive to new research in this direction.
-	The paper is well-written and easy to follow.
-	Supervising experts with semantic losses and obtaining networks specialized in specific semantic areas of the input signal might unlock several applications for INRs and ease their interpretability.
-	The proposed framework achieves good reconstruction performance.
-	The ablations in Table 4 and Table 5 are very insightful.

**Weaknesses:**

W1) The major weakness of the paper is the misalignment between the experimental results and the motivations of this research:

 a- In the introduction (L26-28), the authors correctly point out that, in traditional INRs, each coordinate needs to be processed by the whole network. Even though they claim that this problem can be solved by MoE INRs, in the proposed architecture, the input coordinate needs to be processed by the full manager and the encoder of the expert branch. The only saved computation is the one from the final expert, a much smaller network than the others. Thus, the saved computation looks minimal to standard SIREN (considering the same total weights). Moreover, no experiments regarding computational efficiency and the advantages of parallelized inference are necessary to motivate this claim. Maybe, instead of talking about the absolute efficiency of the proposed approach, it is better to show the better trade-offs in terms of efficiency and reconstruction quality than SIREN.

b- In the introduction (L28-30), the authors claim that standard INRs extract features vastly different for close spatial coordinates (i.e., locality problem). I am unaware of studies that formally investigate this INR property. Thus, I suggest adding a reference work or validating it with experiments. Moreover, the authors claim MoE INRs can learn local piece-wise functions (L259 in the conclusion section). Thus, they do not suffer from the problem above. Yet, the experiments show something different. For instance, by looking at Figure 3c, different experts predict audio signals for close temporal coordinates. I can notice the same behavior in the last column of Figure 6, in which many distinct experts predict pixels in the upper part of the image.

W2 ) The idea of using MoE resembles the idea of KiloNeRF [1]. In that case, the routing strategy is not learned and depends only on the input 3D coordinate, and each expert focuses on a pre-determined spatial region. I think the authors should add this reference, explaining the pros and cons of the two kinds of approaches.

W3) In recent years, INR frameworks have been proposed to be faster and more efficient than MLP-only techniques such as SIREN. For instance, hybrid INRs such as InstantNGP [2] (hash grid + MLP) can be used as the base framework to speed up computation when learning INRs of images and surfaces or NeRFs. The paper should also include more recent competitors than SIREN.


[1] Kilonerf: Speeding up neural radiance fields with thousands of tiny mlps. ICCV 2021.
[2] Instant neural graphics primitives with a multiresolution hash encoding. ACM transactions on graphics (TOG) 2022.

**Questions:**

Q1) In Table 1, does the vanilla MoE baseline employ Softplus activations, while the final strategy uses Sine activations? In this case, the comparison is unfair and does not validate the superiority of their approach to the vanilla one.

Q2) Can the authors also include the Chamfer Distance metric in Table 3?



I like the paper's core idea. However, the author's response to my concerns will greatly influence my final rating.

**Limitations:**

The paper includes a discussion on limitations.

---

> ### Author Rebuttal · Authors · 2024-08-07
>
> W1)a: The limitation we are mainly interested in is learning capacity rather than computation time. For traditional INRs, as each coordinate needs to be processed by the whole network, all parameters have to contribute to the output of every point in the domain, making parameter optimization difficult. For our proposed network, we have subnetworks that are specialised for particular regions of the domain, and thus it is easier to optimise towards better solutions (as our results show). We will make this more clear in our revised version. As the reviewer mentions, saved computational time would be minimal (and highly dependent on hardware architecture), hence we have not explored this.
>
> W1)b: We apologize for the poor wording regarding locality limitation. What we meant is that a desirable property of INRs is for features to change rapidly (to allow modelling of sharp boundaries). This can't be achieved in networks without explicit spectral bias (which we mention in lines 62-66). We will update the paper to clarify this. Our method gives an alternative way for handling sharp changes, i.e., by changing the expert, which is a more natural way to handle discontinuous signals. This is consistent with your observations in Figures 3c and 6.
>
> W2): We will add this reference. Apart from the obvious difference (ours is an MoE so can learn different regions, while they have fixed regions in a grid structure), KiloNeRF's priority is post optimization rendering speed, hence their simple structure. Their architecture leads to insufficient performance when optimizing directly (see their Figure 3), so they rely on distillation to train. Our method on the other hand is an alternative for the original network that optimizes directly with better signal reconstruction performance, which is not the case for KiloNeRF. Furthermore we show improvement over grid based methods in Table 5.
>
> W3): See general comment. We leave combining our MoE approach with hash based methods as future work. Also note that our updated table 1 has comparision with FINER from CVPR2024 (see general comment).
>
> Q1): No, vanilla MoE was with Sine activations, we will change the name to Vanilla SIREN MoE to make it more clear
>
> Q2): We have added the Chamfer distance now (see table in general comment).

---

> > ### Comment · Area_Chair_2LXj · 2024-08-12
> >
> > Regarding W3, could you provide the results using InstantNGP as the reviewer VtKc requested? Considering the potential impact of this work, we look forward to seeing this comparison.

---

> ### Author Response · Authors · 2024-08-13
>
> **Results with InstantNGP**: We give results with InstantNGP in the updated Table 1 (put as a new general comment above). The results show that our Neural Experts SIREN and Neural Experts FINER achieve superior performance compared to InstantNGP (89.35 and 90.84 PSNR vs 84.56 PSNR) with an order of magnitude fewer parameters (366k vs 7.7M). In the InstantNGP experiment, we use their default architecture for image experiments (16 encoding levels, 2 parameters per level, maximum cache size of 2^19, and a 2 hidden layer decoder with 64 neurons). Note that most of InstantNGP's parameters are spatial parameters (hash encoding), specifically 99.86% of all parameters. This comparison between our Neural Experts SIREN/FINER and InstantNGP shows that the spatial parameters (parameters dedicated for specific spatial regions, so the experts and the hash encoding) are effective, but if used with patterns and heuristics (like InstantNGP or the other methods discussed in lines 70-76) many of the spatial parameters become redundant and underutilized quickly. This motivates the learning of the spatial regions to achieve a parameter efficient approach (our Neural Experts method).
>
> Extending InstantNGP to include the MoE architecture is non-trivial as the hash encoding parameters, which make up over 99% of the total parameters, are used in the beginning of the network (to provide a specialized encoding to then go through a tiny MLP decoder). The obvious Neural Experts extension (which requires a shared input encoding), is to apply MoE to that tiny decoder, which we call Naive Neural Experts InstantNGP. Results are reported in the updated table 1, and we observe that the baseline InstantNGP performs better than this variant. However, the shared input hash encoding parameters are still over 99% of the total parameters. As shown in our response to Reviewer ekS9, the parameter allocation ratio between the input encoder, the experts and the manager plays an important role in the MoE's performance, with the best performing ratio being 14%, 55% and 31% respectively. So this naive result is to be expected. Constructing a more fair distribution of parameters would require a fundamental change to the InstantNGP backbone. Either we greatly increase the parameters of the expert and manager (which would make the total parameters another order of magnitude larger) or we need a way to split the hash encoding parameters into the experts. Both of these extensions are outside the scope of the current work.
>
> Overall, these results show that learning regions for spatial parameters, as in our Neural Experts, achieve better performance with significantly fewer parameters.

---

> > ### Comment · Reviewer_VtKc · 2024-08-13
> >
> > I thank the authors for the responses. I appreciated the experiments with InstantNGP. Most of my concerns are addressed. Thus, I will raise my score to Weak Accept (6).
> >
> > I expect the authors to insert the new results with InstantNGP and update the introduction to align more with the considerations in W1a and W1b.

---

### Official Review · Reviewer_aVob · 2024-07-15

**Soundness:** 3
**Presentation:** 3
**Contribution:** 3
**Rating:** 6
**Confidence:** 3

**Summary:**

This paper introduces a MoE architecture for INRs, enhancing scalability, local fitting, and parallelization. Traditional INRs use a single network, imposing global constraints, while this method learns several local expert functions, subdividing the domain. Key contributions include a novel manager architecture and initialization method for better convergence and domain subdivision without ground truth. It show improved performance and efficiency over traditional methods, across image, audio, and 3D surface reconstruction,

**Strengths:**

- It demonstrates a neat architectural design along with a robust ablation study.
- It consistently shows performance improvement across various tasks, and the performance with respect to the number of parameters is also superior.
- It is interesting that random initialization, which includes no inductive bias in the manager pretraining process, outperforms initializations like SAM.

**Weaknesses:**

- It requires more parameters compared to the vanilla model. While it is fair to compare it with the wider version of the vanilla model, it is unclear if the proposed model still performs well when it has the same number of parameters as the vanilla model (not wider version). Tab.3 alleviates this concern to some extent.

- In addition to comparing with the vanilla model, it would be good to include a discussion on various INR methods that apply locality bias (e.g., spatial functa).

- As shown in the convergence graphs in the appendix, it shows more unstable convergence compared to the baseline.

**Questions:**

- Please see the above weaknesses.
- In Tab.1, what is the PSNR of Neural Experts SIREN with the same number of parameters as SIREN?

**Limitations:**

The authors have adequately addressed the limitations.

---

> ### Author Rebuttal · Authors · 2024-08-07
>
> W1: We have added results in the updated Table 1 in General Comments (see "Ours Neural Experts SIREN small **(New)**"). This version has a width of 68 in each layer (encoding, experts and manager) instead of 128, leading to 98,686 parameters. Note that it still outperforms the SIREN baseline, which has 99k parameters.
>
> W2: See comment in General Comments.
>
> W3: Yes, the convergence is unstable as we optimizing both the parameters and the regions. However our results show that even with unstable convergence we have a consistent performance increase from our baselines, and thus show the promise of this line of work.

---

> > ### Comment · Reviewer_aVob · 2024-08-10
> >
> > Thank you for the authors' response. After reading all the reviews and their responses, I found several potential weaknesses I had not noticed. But I think the authors addressed most of the major things pointed out. Their new experimental results addressed my concerns and the contribution about 'benefit of optimizing the regions and how to make this work in practice' is convincing to me.
> >
> > I agree that recent INR methods that apply locality bias mentioned in the reviews have different main goals, but I'm still unsure if this method can have a meaningful impact on INR-based applications without such comparisons.
> >
> > Therefore, I'd like to maintain my rating of 'weak accept'.

---

### Author Rebuttal · Authors · 2024-08-06

## General Comments
We thank the reviewers for their insightful comments. We include requested additional experimental results here and response to common questions. Specific questions are addressed to each reviewer below.

**New Table 1** (page 5 of submission) now updated to have comparison with Softplus + FF (Fourier features [38]) based networks and FINER based networks [1, new], a version of our method with similar parameter count to SIREN, and SSIM and LPIPS metrics.

| **Method**| **Parameters** | **PSNR Mean**| **PSNR Std**| **SSIM Mean**| **SSIM Std**| **LPIPS Mean** | **LPIPS Std** |
|---------------------------------------------|----------------|----------------|---------------|----------------|---------------|----------------|---------------|
| Softplus| 99.8k| 19.51| 2.95| 0.7158 | 0.1106| 4.08e-1| 8.33e-2 |
| Softplus Wider| 642.8k | 20.91| 3.12| 0.7798 | 0.0899| 3.38e-1| 8.44e-2 |
| Ours Neural Experts SoftPlus| 366.0k | 20.62| 3.12| 0.7628 | 0.0946| 3.53e-1| 8.47e-2 |
| Softplus + FF **(New)** | 99.8k| 28.97| 3.30| 0.9433 | 0.0193| 7.59e-2| 1.70e-2 |
| Softplus + FF Wider **(New)** | 642.8k | 29.48| 3.91| 0.9436 | 0.0245| 7.74e-2| 2.30e-2 |
| Ours Neural Experts SoftPlus + FF **(New)** | 366.0k | 31.66| 3.16| 0.9652 | 0.0180| 4.13e-2| 1.57e-2 |
| SIREN | 99.8k| 57.23| 2.46| 0.9991 | 0.0005| 5.78e-4| 5.09e-4 |
| SIREN Wider | 642.8k | 77.50| 5.32| 0.9996 | 0.0005| 3.08e-4| 3.04e-4 |
| Vanilla SIREN MOE | 349.6k | 62.98| 4.16| 0.9993 | 0.0005| 4.53e-4| 3.88e-4 |
| Ours Neural Experts SIREN small **(New)** | 98.7k| 63.42| 7.09| 0.9992 | 0.0007| 9.96e-4| 2.40e-3 |
| Ours Neural Experts SIREN | 366.0k | 89.35| 7.10| **0.9997** | 0.0004| 2.49e-4| 2.93e-4 |
| FINER **(New)** | 99.8k| 58.08| 3.04| 0.9991 | 0.0005| 7.47e-4| 1.31e-3 |
| FINER Wider **(New)** | 642.8k | 80.32| 5.40| 0.9996 | 0.0004| 2.49e-4| 2.46e-4 |
| Ours Neural Experts FINER **(New)** | 366.0k | **90.84**| 8.14| **0.9997** | 0.0004| **2.46e-4**| 2.48e-4 |
Note that for the new results, our method outperforms the baselines.

1 (new). Liu, Zhen, et al. "FINER: Flexible spectral-bias tuning in Implicit NEural Representation by Variable-periodic Activation Functions." Proceedings of the IEEE/CVF Conference on Computer Vision and Pattern Recognition. 2024.

**Table 3** We also add Chamfer results to Table 3 (page 7 of the submission).

| **Method**                    | **# params** | **Armadillo**   | **Dragon**      | **Lucy**        | **Thai Statue** | **Mean**        |
|-------------------------------|--------------|-----------------|-----------------|-----------------|-----------------|-----------------|
| SIREN Large                   | 1.5M         | 1.4983e-05      | 2.2367e-05      | 1.2030e-04      | 5.8465e-05      | 5.4029e-05      |
| Our Neural Experts Large      | **1.3M**     | **1.4975e-05**  | **2.1340e-05**  | **1.1322e-04**  | **5.4084e-05**  | **5.0905e-05**  |
| SIREN Small                   | 396k         | 2.9165e-05      | 2.3944e-05      | 1.2583e-04      | 6.7273e-05      | 6.1553e-05      |
| Our Neural Experts Small      | **323k**     | **1.5086e-05**  | **2.2102e-05**  | **1.1047e-04**  | **5.9721e-05**  | **5.1845e-05**  |

Consistent with Trimap IoU (reported in the submission), our method outperforms the baseline on Chamfer distance.

**Discussion about newer INR methods that apply locality bias (such as hybrid methods)**: We discuss such methods in lines 70-76. These methods greatly improve signal reconstruction performance and often also faster as reviewer VtKc mentions. However, as they choose regions in a deterministic or heuristical way [7,15,24,27,33,37], in order to improve task importance many regions are required, leading to these methods being very parameter inefficient (see InstantNGP [27] Fig 2 and 4). Rather than looking into ways to make more efficient patterns or heruistics for assigning spatial parameters, we consider fixing the number of experts (region specific parameter sets) and allowing their region of influence to be optimized, so that for a smaller number of parameters we can get better performance. Thus, while our method is of the same category as those methods (specializing parameters spatially) our method has a different goal: to show the benefit of optimizing the regions and how to make this work in practice. In fact we show that what is required to make this work can be counterintuitive (needs encoder and random pretraining) so we believe this work is of great interest to the community.

**Using our approach with NeRFs**
Extending our MoE method to NeRFs is an intersting future direction as it cannot natively fit within an MoE framework. In an MoE framework, we change a loss per instance $\mathcal{L}_i=\ell(f(x_i),y_i)$ to a weighted sum of losses over experts $\mathcal{L}_i=\sum_j q_j \ell(f_j(x_i),y_i)$ where $f$ is our model, $f_j$ are the expert models, $q_j$ is the expert weights, and $x_i,y_i$ are the current instance's input and ground truth. Thus as reviewer ekS9 points out, we need a 3D ground truth signal. However NeRF only has supervision per ray (color for each pixel), and the rendering model accumulates points along a ray. Thus there is no way to specify a loss per expert, as the points along a ray may be assigned to different experts. While we could then specialise experts for each view direction, we cannot specialise experts for each 3D point with this formulation. This may be addressed by modifying the formulation, e.g., through a variational bound, but is beyond the scope of the current work.

---

### Author Response · Authors · 2024-08-13
**New results following discussion from Reviewer VtKc (W3)/Area chair 2LXj, and Reviewer ekS9**

**New Table 1** (page 5 of submission, updated for the second time) now updated to have comparison with InstantNGP as the base architecture and with random pretraining the Vanilla SIREN MOE.

| **Method**| **Parameters** | **PSNR Mean**| **PSNR Std**| **SSIM Mean**| **SSIM Std**| **LPIPS Mean** | **LPIPS Std** |
|-|-|-|-|-|-|-|-|
| Softplus| 99.8k| 19.51| 2.95| 0.7158 | 0.1106| 4.08e-1| 8.33e-2 |
| Softplus Wider| 642.8k | 20.91| 3.12| 0.7798 | 0.0899| 3.38e-1| 8.44e-2 |
| Ours Neural Experts SoftPlus| 366.0k | 20.62| 3.12| 0.7628 | 0.0946| 3.53e-1| 8.47e-2 |
| Softplus + FF | 99.8k| 28.97| 3.30| 0.9433 | 0.0193| 7.59e-2| 1.70e-2 |
| Softplus + FF Wider | 642.8k | 29.48| 3.91| 0.9436 | 0.0245| 7.74e-2| 2.30e-2 |
| Ours Neural Experts SoftPlus + FF | 366.0k | 31.66| 3.16| 0.9652 | 0.0180| 4.13e-2| 1.57e-2 |
| SIREN | 99.8k| 57.23| 2.46| 0.9991 | 0.0005| 5.78e-4| 5.09e-4 |
| SIREN Wider | 642.8k | 77.50| 5.32| 0.9996 | 0.0005| 3.08e-4| 3.04e-4 |
| Vanilla SIREN MOE | 349.6k | 62.98| 4.16| 0.9993 | 0.0005| 4.53e-4| 3.88e-4 |
| Vanilla SIREN MoE + Random Pretraining **(New)** | 349.6k | 74.28 | 7.36| 0.9996 | 0.0004 | 3.05e-4 | 2.88e-4 |
| Ours Neural Experts SIREN small | 98.7k| 63.42| 7.09| 0.9992 | 0.0007| 9.96e-4| 2.40e-3 |
| Ours Neural Experts SIREN | 366.0k | 89.35| 7.10| **0.9997** | 0.0004| 2.49e-4| 2.93e-4 |
| FINER | 99.8k| 58.08| 3.04| 0.9991 | 0.0005| 7.47e-4| 1.31e-3 |
| FINER Wider | 642.8k | 80.32| 5.40| 0.9996 | 0.0004| 2.49e-4| 2.46e-4 |
| Ours Neural Experts FINER | 366.0k | **90.84**| 8.14| **0.9997** | 0.0004| **2.46e-4**| 2.48e-4 |
| InstantNGP **(New)** | 7.7M | 84.56 | 5.62| 0.9996 | 0.0003 | 4.28e-4 | 5.37e-4 |
| Naive Neural Experts InstantNGP **(New)** | 7.7M | 75.14 | 2.57| 0.9996 | 0.0004 | 4.07e-4 | 4.35e-4 |

---

### Decision · Program_Chairs · 2024-09-25

**Decision:**

Accept (poster)

**Comment:**

**Summary:**
Implicit neural representations (INRs) excel in tasks like image, shape, audio, and video reconstruction but usually rely on a single network, imposing global constraints. Contrary to the traditional, this paper introduced a mixture of experts (MoE), enabling local, piece-wise learning that improves domain fitting. Integrating MoE into INR frameworks boosts speed, accuracy, and memory efficiency, with new methods for gating network conditioning and pretraining further enhancing performance. The research direction is both intriguing and novel, making it well-suited for presentation at the conference.

**Strengths:** may include:
- "Neat architectural design" (aVob)
- Strong ablation study (aVob, VtKc, ij4R)
- "Consistently" outperformance (aVob)
- "Well-written and easy to follow" (VtKc, ij4R, ekS9)
- "a novel MoE-based architecture," "is novel" (ij4R, ekS9)

**Weaknesses:**
Initially, reviewers raised concerns about having "more parameters" than the baselines (aVob), the need for further discussion on comparative methods (e.g., spatial functa, KiloNeRF, InstantNGP, Gaussian Activation Function, WIRE, FINER) by the Reviewers aVob, VtKc, ij4R, and ekS9, "unstable convergence" (aVob), and "misalignment between the experimental results and the motivations" (VtKc).

**Justifications for Acceptance:**
However, after the rebuttal and discussions with the reviewers, the authors provided experimental evidence and explanations for their decisions, which led to acceptance from all the reviewers. Especially, AC appreciates the authors' efforts and quick response to the request for additional evidence demonstrating the broad impact of their work by providing InstantNGP results, which sufficiently resolved the major issues raised. Based on this, AC is pleased to recommend acceptance.